# Inhibition of the Niemann-Pick C1 protein is a conserved feature of multiple strains of pathogenic mycobacteria

Yuzhe Weng [1,4], Dawn Shepherd[1,4], Yi Liu [2], Nitya Krishnan[3], Brian D. Robertson[3], Nick Platt[1,5], Gerald Larrouy-Maumus[2,5] & Frances M. Platt [1,5] ✉

*Mycobacterium tuberculosis* (*Mtb*) survives and replicates within host macrophages (MΦ) and subverts multiple antimicrobial defense mechanisms. Previously, we reported that lipids shed by pathogenic mycobacteria inhibit NPC1, the lysosomal membrane protein deficient in the lysosomal storage disorder Niemann-Pick disease type C (NPC). Inhibition of NPC1 leads to a drop in lysosomal calcium levels, blocking phagosome-lysosome fusion leading to mycobacterial survival. We speculated that the production of specific cell wall lipid(s) that inhibit NPC1 could have been a critical step in the evolution of pathogenicity. We therefore investigated whether lipid extracts from clinical *Mtb* strains from multiple *Mtb* lineages, *Mtb* complex (MTBC) members and non-tubercular mycobacteria (NTM) inhibit the NPC pathway. We report that inhibition of the NPC pathway was present in all clinical isolates from *Mtb* lineages 1, 2, 3 and 4, *Mycobacterium bovis* and the NTM, *Mycobacterium abscessus* and *Mycobacterium avium*. However, lipid extract from *Mycobacterium canettii*, which is considered to resemble the common ancestor of the MTBC did not inhibit the NPC1 pathway. We conclude that the evolution of NPC1 inhibitory mycobacterial cell wall lipids evolved early and post divergence from *Mycobacterium canettii*-related mycobacteria and that this activity contributes significantly to the promotion of disease.

*Mycobacterium tuberculosis* (Mtb), the causative agent of tuberculosis (TB) is a major human pathogen[1,2]. While the emergence of multi-drug-resistant strains has added to the challenges of treatment, the ability of Mtb to survive and replicate intracellularly in host macrophages (MΦ) is key to its success[3,4]. Typically, bacilli within aerosol droplets are inhaled into the lower pulmonary environment and initially infect resident alveolar macrophages (MΦ), which induces the recruitment of additional myeloid cells, leading to the formation of granulomas. Granulomas facilitate infection of other cell types and promote

mycobacterial growth[3,5]. Long-term intracellular persistence is the basis for TB latency, which provides a pathogen reservoir that can subsequently become reactivated and promote transmission to a new host[6]. Mycobacteria are phagocytosed following an initial interaction with MΦ plasma membrane receptors[7,8]. However, in contrast to the fate of other ingested particles, which involves phagosome formation and subsequent phagosome-lysosome fusion, Mtb and other pathogenic mycobacteria actively block this process facilitating their long-term survival within the MΦ[5,9]. Although multiple mechanisms have

[1]Department of Pharmacology, University of Oxford, Mansfield Road, Oxford OX1 3QT, UK. [2]MRC Centre for Molecular Bacteriology and Infection, Department of Life Sciences, Faculty of Natural Sciences, Imperial College London, London, UK. [3]MRC Centre for Molecular Bacteriology and Infection, Department of Infectious Disease, Imperial College London, Flowers Building, London SW7 2AZ, UK. [4]These authors contributed equally: Yuzhe Weng, Dawn Shepherd. [5]These authors jointly supervised this work: Nick Platt, Gerald Larrouy-Maumus, Frances M. Platt. ✉e-mail: frances.platt@pharm.ox.ac.uk

been proposed to explain the ability of Mtb to inhibit the fusion of the lysosome with the phagosomes[10,11] current understanding remains incomplete. The possibility of interfering with the Mtb-mediated block in phagosome-lysosome fusion is an attractive target for host-directed Mtb therapy.

As well as the block in phagosome–lysosome fusion, cells harboring intracellular mycobacteria display several additional characteristics[5] and we have previously reported the unexpected similarity between the cellular phenotypes of Mtb-infected MΦ and those of the rare, inherited lysosomal storage disease, Niemann-Pick disease type C (NPC)[12]. NPC is caused by mutations in either of two genes, *NPC1* (95% of clinical cases) or *NPC2*. They work cooperatively in a pathway that is involved in lipid trafficking and lysosome: ER contact site formation[13,14]. The precise function of NPC1 is incompletely understood and a direct assay to measure its function is therefore not available. We therefore demonstrated that cells harboring specific mycobacteria display the unique combination of cellular phenotypes that result from loss of NPC1 activity[12]. In brief, all of the cellular phenotypes that define NPC, including a reduction in the calcium content of lysosomes, prevention of phagosome-lysosome fusion, accumulation of specific lipid species (including cholesterol, sphingomyelin, and glycosphingolipids (GSLs)) within the endo-lysosomal system were induced in cells infected with mycobacteria that survive within host cells (Mtb and *Mycobacterium bovis Bacillus* Calmette-Guerin (BCG)), but not the environmental non-pathogenic mycobacterium *Mycobacterium smegmatis*. Very significantly, it was the lipid fraction of the cell wall from the pathogenic mycobacteria that inhibited the NPC pathway. Critically, all of the NPC cellular phenotypes were induced not only in infected and lipid-treated murine RAW 264.7 MΦ but also in human, monocyte-derived MΦ, thereby confirming relevance to human disease[12]. Additionally, we observed induction of the NPC cellular phenotypes in neighboring cells that did not contain internalized mycobacteria. These data are consistent with the proposal that pathogenic mycobacteria shed lipids that inhibit the activity of NPC1, and thereby prevent normal phagosome–lysosome maturation[12].

We hypothesized that the evolution of lipids that inhibit NPC1 would correlate with the acquisition of the ability to survive intracellularly and cause disease. In this study, we have therefore assayed the capacity of cell wall lipid extracts to induce NPC cellular phenotypes in RAW 264.7 MΦ. We report that extracts from diverse pathogenic mycobacterial species representing the main Mtb lineages differentially induce NPC phenotypes with the notable exception of *Mycobacterium canettii*, which is believed to resemble the putative common ancestor. These findings suggest that the evolution of lipids capable of inhibiting NPC1 evolved early, post-divergence from *M. canettii* and correlates with virulence and intracellular survival and therefore represents a new target for therapy.

## Results

### Cell wall lipid fractions from Mtb clinical isolates, specific NTM, and *M. bovis Bacillus* but not *M. canettii* increase LysoTracker™ staining intensity of RAW 264.7 MΦ

We have reported previously that the volume of the acidic compartment (late endosome/lysosome (LE/Lys)) is significantly expanded in genetic NPC1-deficient cells[15] and in RAW 264.7 MΦ treated with the cationic amphiphile U18666A, which binds to NPC1 and inhibits its activities[16] as well as in cells infected with BCG[12]. Using the fluorescent probe LysoTracker™, which selectively accumulates in the acidic compartment, we measured by FACS the relative LE/Lys volume in cells that had been incubated with lipids at a concentration of 100 μg/ml for 48 h. We assayed lipids extracts prepared from mycobacterial clinical strains representative of the major Mtb lineages (Table 1) and other mycobacterial species (Supplementary Fig. 1). Their activities were compared with vehicle-treated (negative control) and U18666A-incubated (positive control) RAW 264.7 MΦ (FACS gating strategy; Supplementary Fig. 2). All of the other strains tested induced a statistically significant increase in LE/Lys volume as shown by greater LysoTracker™ staining (Fig. 1a) except for strains 232 and 346 from lineage 1 and 212 and 374 from lineage 2, which showed a similar trend. Strains 119, 173, 293, 367, 440, 369, and H37RV increased LysoTracker™ staining to a level that was not significantly lower than that of cells treated with the NPC1 pharmacological inhibitor, U18666A. Analysis of the data for each Mtb lineage confirmed there was a significant increase in LysoTracker™ staining of RAW 264.7 MΦ incubated with lipid extracts of all lineages (Fig. 1b).

Analysis of lipids from other mycobacterial species revealed that the MTBC member *M. bovis Bacillus* and NTM species *M. abscessus* and *M. avium* also significantly increased LysoTracker™ staining, while extracts from *M. smegmatis*, the non-pathogenic model organism did not achieve significance and *M. canettii* diminished reduced LysoTracker™ staining relative to control cells (Fig. 1a).

At a concentration of 100 μg/ml, we were unable to measure a significant increase in LysoTracker™ staining of RAW 264.7 cells following exposure to lipid extracts from two Mtb strains within lineage 1 (isolates 232 and 346) and two strains from lineage 2 (isolates 212 and 374). We reasoned that this could perhaps be due to lower concentrations of the active lipid species in these extracts. We therefore performed repeat assays but increased the concentration of lipid extracts to 200 μg/ml with which we observed a statistically significant increase in LysoTracker™ with all four strains (Fig. 1c).

Furthermore, even at the higher lipid concentration of 200 μg/ml *M. canettii* had no effect, whereas the Mtb strain, 91_0079, a representative of lineage 3 tested at the same concentration significantly enhanced staining, thereby demonstrating the presence of activity within this lineage (Fig. 1c).

Mycobacterial lipid extracts isolated at two separate laboratories were also assayed; one from Imperial College, London (ICL) and the second from Colorado State University, supplied through BEI Resources, Virginia. In order to exclude the possibility of significant variation in activity due to methodological differences in their isolation or bacterial growth conditions, we compared lipid extracts of H37Rv from each source within the same experimental assay. The two

## Table 1 | Genotypes of the selected Mtb clinical isolates strains used in this study

| Strain | Spoligotype[I] | ST number[II] | Lineage category[III] | Lineage number[IV] | Lineage geographical location[V] |
|--------|-----------|----------|------------------|----------------|---------------------------|
| 232 | EA14_VNM | ST139 | Ancient | 1 | IO |
| 346 | EA14_VNM | ST140 | Ancient | 1 | IO |
| 372 | EA14_VNM | ST141 | Ancient | 1 | IO |
| 281 | ZERO | ST405 | Ancient | 1 | IO |
| 374 | BEIJING | ST1 | Modern | 2 | BJ |
| 212 | BEIJING | ST2 | Modern | 2 | BJ |
| 333 | BEIJING | ST3 | Modern | 2 | BJ |
| 119 | BEIJING | ST4 | Modern | 2 | BJ |
| 345 | BEIJING LIKE | Unclassified | Modern | 2 | BJ |
| 649 | BEIJING LIKE | ST269 | Modern | 2 | BJ |
| 318 | T1 | ST53 | Modern | 4 | EA |
| 173 | H1 | ST62 | Modern | 4 | EA |
| 367 | ZERO | ST405 | Ancient | 1 | IO |
| 440 | H3 | ST946 | Modern | 4 | EA |
| 639 | H3 | ST50 | Modern | 4 | EA |
| 293 | T1 | ST53 | Modern | 4 | EA |

I and II Classified in accordance with the international Spoligotyping database[78]. III Classified based on evolutionary phylogenetic tree[79]. IV Classified in accordance with sequence alignment[36]. V. *BJ*. East Asian/Beijing; *IO*. Indo-Oceanic; *EA*. Euro-American.

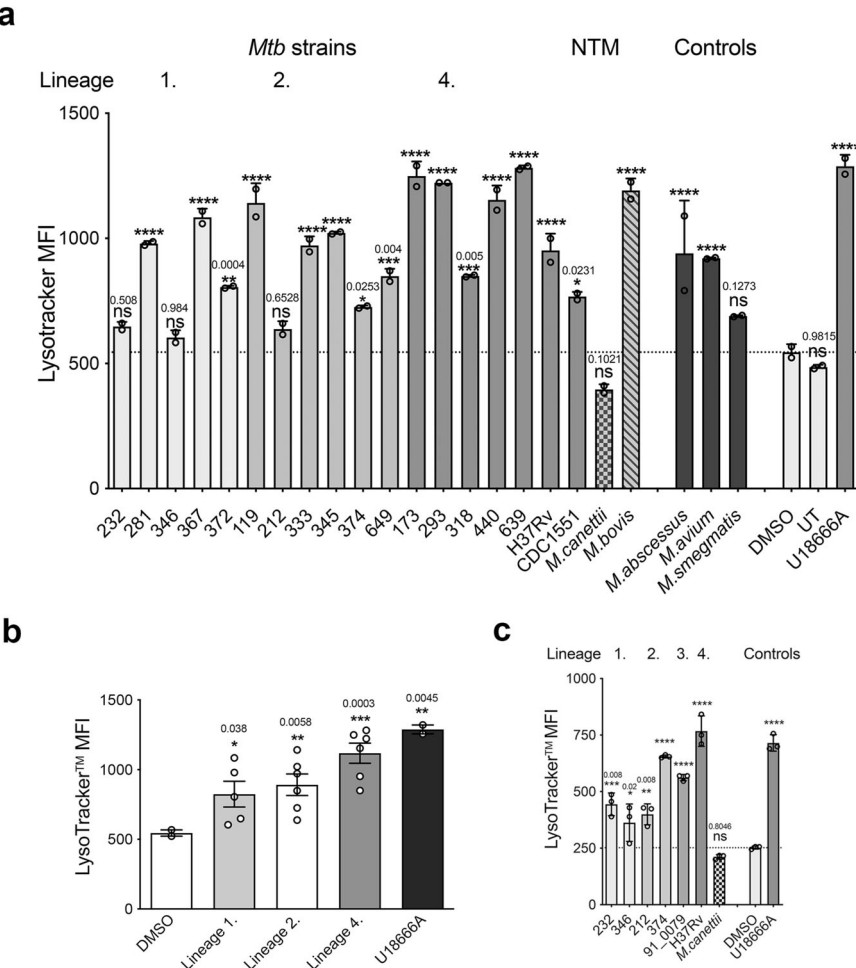

**Fig. 1 | Mycobacterial lipids that can significantly increase LysoTracker™ staining of RAW 264.7 MΦ are widely distributed across Mtb clinical strains, members of the Mtb complex (MTBC) and non-tuberculous mycobacteria.**
**a** LysoTracker™ staining intensity of RAW 264.7 MΦ treated with 100 μg/ml lipid extract of clinical Mtb strains representative of Mtb lineages, 1, 2, and 4, Mtb complex species and non-tuberculous mycobacteria for 48 h. Data are mean ± SEM, $N$ = minimum of 2 replicates per sample. Statistical analysis, 1-way ANOVA, ****$p$ <0.0001, other significance values as indicated. Data are representative of three independent experiments. **b** Mean LysoTracker™ staining intensity of RAW

264.7 MΦ treated with lipid from Mtb lineages, 1, 2, and 4. Data are mean ± SEM, $N$ = 5–7 strains tested per lineage. Statistical analysis, 1-way ANOVA, ****$p$ < 0.0001, other $p$ values as indicated. Data are representative of three independent experiments. **c** LysoTracker™ staining of RAW 264.7 MΦ treated with 200 μg/ml lipid extract of clinical Mtb strains from lineages 1, 2, 3, and 4 and *M. canettii*. Data are mean ± SEM, $N$ = minimum of 3 replicates per sample. Statistical analysis, 1-way ANOVA, ****$p$ < 0.0001, other $p$ values as indicated. Data are representative of two independent experiments. Source data are provided as a Source data file.

extracts significantly increased LysoTracker™ staining to the same extent, which was equal to or greater than that of U18666A-treated MΦ (Supplementary Fig. 3a). Furthermore, we confirmed dose-dependent responses for both H37Rv samples (Supplementary Fig. 3a) and for lipid extracts of several of the positive Mtb clinical strains (Supplementary Fig. 3b). We also tested the ability of specific non-lipid components of Mtb to increase LysoTracker™ staining and found that mycolylarabinogalactan-peptidoglycan (mAGP), lipomannan (LM) lipoarabinomannan (LAM) and arabinogalactan (all from BEI) did not have a significant effect on staining intensity (Supplementary Fig. 3c).

**Cell wall lipid fractions from representative Mtb clinical isolates but not *M. canettii* increase lysosome number and size in treated RAW 264.7 MΦ**
The intensity of LysoTracker™ staining when measured by FACS provides a measure of the relative acidic compartment volume in the cell[15]. This could represent an increase in the number and/or size of lysosomes per cell. We therefore stained cells that had been incubated with lipid extracts of Mtb clinical stains representative of three lineages (lineage 1, strain 281; lineage 2, strain 333; lineage 4, strains 639 and

H37Rv) and *M. canettii*, which did not induce an increase in Lyso-Tracker™ intensity for examination by confocal microscopy (Fig. 2a). Analyses of the frequency of lysosomes, as indicated by the distribution of fluorescence confirmed that there was a statistically significant greater number of LysoTracker™ stained organelles in cells treated with U18666A and the lipid extracts of each of the Mtb strains, but not in RAW 264.7 MΦ exposed to *M. canettii* lipid (Fig. 2b). We measured the diameter of the fluorescent puncta and determined that there was a significant increase in the mean size and size distribution with U18666A (Fig. 2c) and Mtb lipid (Fig. 2d) treatment, but not with *M. canettii* (Fig. 2d). These data indicate that Mtb lipids increase both the number and size of lysosomes but *M. canettii* does not affect either parameter.

**LAMP-1 staining confirms expansion of late endosome/lysosomal volume following incubation with lipids from representative Mtb strains but not *M. canettii*, but its level of expression is not significantly increased**
In order to further validate the expansion of the acidic compartment (LE/Lys), we imaged RAW 264.7 MΦ that had been treated with

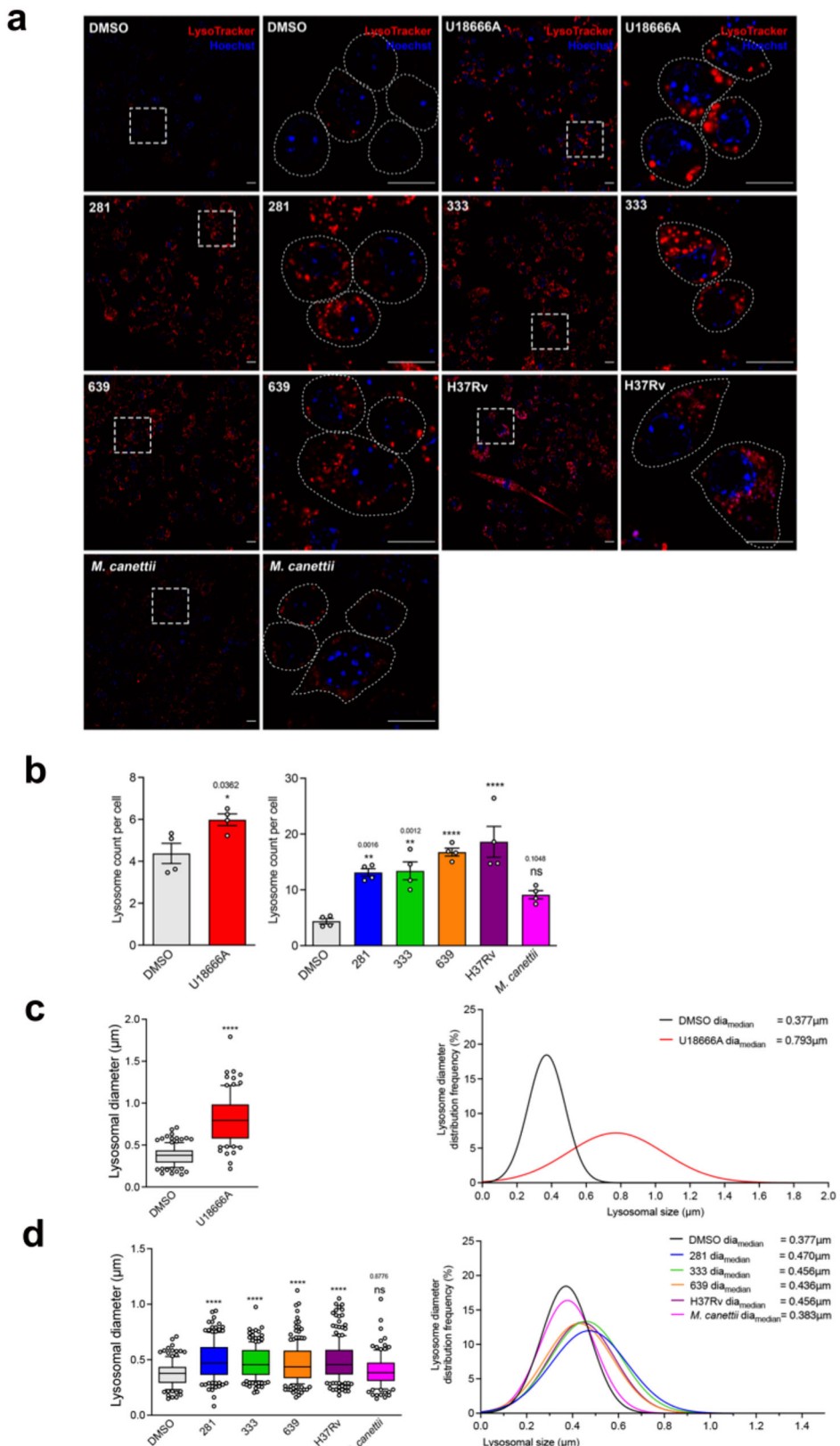

U18666A or lipid from the specific Mtb strains and stained for the protein LAMP-1 (Fig. 3a, b). Quantification of LAMP-1 positive compartments confirmed a significant increase in the mean size (diameter) and size distribution of LAMP-1 positive compartments of cells treated with U18666A (Fig. 3c) and Mtb lipids, but not *M. canettii* (Fig. 3d).

However, Q-PCR analysis (Fig. 3e) and western blotting (Fig. 3f) did not confirm significant increases in either LAMP-1 transcription or protein abundance in Mtb H37Rv lipid treated cells, implying that differences in confocal images likely reflect a change to protein distribution affecting immunoreactivity.

**Fig. 2 | Expansion of the lysosomal compartment in RAW 264.7 MΦ in response to treatment with mycobacterial lipid (100 µg/ml for 48 h) visualized with LysoTracker™-Red staining. a** Representative confocal microscopy images of RAW 264.7 MΦ either untreated, DMSO or U18666A treated or incubated with lipid extracts of Mtb clinical strains 281, 333, 639 or H37Rv or *M. canettii* stained with LysoTracker™-Red. The right image of each pair is a higher magnification of the region indicated in the left image. Nucleus is stained with Hoechst 33342 (blue). Scale bar represents 10 µm. Exposure times for all images was equivalent, except for DMSO indicated as a longer exposure. **b** Quantification of lysosome frequency per cell in vehicle or U18666A (left histogram) or DMSO and Mtb lipid or *M. canettii* lipid treated cells (right histogram). Data are means ± SEM, *N* = 4 replicates per sample comprising a minimum of 50 cells per random field. Statistical analysis, unpaired *t* test with Welch's correction for DMSO and U18666A, two-tailed; 1-way ANOVA with Dunnett test, *F* = 1.945 for DMSO or lipid treated. ****p* < 0.0001, other *p* values as indicated. Data are representative of two independent experiments. **c** Box-and-whisker plot of lysosome diameters (µm) and statistical distribution of lysosome diameters in vehicle or U18666A treated RAW 264.7 MΦ. Left histogram,

data are means ± SEM, *N* = minimum of 100 lysosomes within four random fields. DMSO: min: 0.149 µm, max: 0.709 µm, center: 0.377 µm; U18666A: min: 0.216 µm, max:1.791 µm, center: 0.793 µm. Box contains 90% of all events. Statistical analysis, unpaired *t* test with Welch's correction, two-tailed. ****p* < 0.0001. Statistical distribution values indicated are median diameter (µm). **d** Box-and-whisker plot of lysosome diameter (µm) and statistical distribution of lysosome diameters in DMSO or lipid treated RAW 264.7 MΦ. Data are means ± SEM, *N* = minimum of 100 lysosomes within four random fields. DMSO: min: 0.149 µm, max: 0.709 µm, center: 0.377 µm; 281: min: 0.081 µm, max: 0.941 µm, center: 0.470 µm; 333: min: 0.204 µµm, max: 0.975 µm, center: 0.456 µm; 639: min: 0.161 µm, max: 1.124 µm, center: 0.436 µm; H37Rv: min: 0.180 µm, max: 1.052 µm, center: 0.456 µm; *M. canettii*: min: 0.149 µm, max: 1.047 µm, center: 0.383 µm. Box contains 90% of all events. Statistical analysis, 1-way ANOVA with Kruskal–Wallis test, Kruskal–Wallis statistics: 79.61. ****p* < 0.0001, other *p* values as indicated. Statistical distribution values indicated are median diameter (µm). Data are representative of two independent experiments. Source data are provided as a Source data file.

## H37Rv lipid extract significantly increases Npc1 expression in treated RAW 264.7 MΦ

In our previous study demonstrating the induction of NPC cellular phenotypes in RAW 264.7 MΦ infected with pathogenic mycobacteria, we observed a significant increase in the expression of NPC1[12]. We examined whether treatment with mycobacterial lipids might have the same effect. While U18666A (Fig. 4a) and H37Rv lipid significantly increased the level of NPC1 protein, we were unable to detect an increase with the other Mtb lipids that were tested (Fig. 4b).

## Cell wall lipid extracts from representative Mtb clinical strains and H37Rv but not *M. canettii* induce cholesterol accumulation in the endo-lysosomal system of RAW 264.7 MΦ

Lysosomal storage of cholesterol is a hallmark of NPC and has historically been used for clinical diagnosis[14]. Mtb is a pathogen that manipulates host cell lipid metabolism and transforms infected cells into lipid-laden, foamy MΦ that are enriched in cholesterol and indeed cholesterol is important for the formation of granulomas[5,17,18]. Cholesterol has also been shown to be a critical energy source for Mtb and is essential for survival[19]. We therefore investigated the capacity of mycobacterial lipid extracts to cause cholesterol accumulation. Evidence of cholesterol accumulation was investigated by filipin staining of fixed cells and biochemical measurement of cell extracts using an Amplex Red assay. As expected, imaging of cells that had been treated with the NPC1 inhibitor U18666A confirmed significant lysosomal accumulation of filipin-positive puncta (Fig. 5a), which were seen in almost all MΦ (Fig. 5b). Comparable analysis of cells that had been treated with lipid extracts of the clinical strains 281, 333, 639, and H37Rv also revealed lysosomal puncta (Fig. 5c), the frequency of which were statistically greater than control (Fig. 5d). In contrast, relatively few puncta were detected in RAW 264.7 cells treated with an extract of *M. canettii* (Fig. 5a) and were not significantly greater in number than control RAW 264.7 cells (Fig. 5d). We measured the total cholesterol content of treated cells. Cells treated with U18666A had a statistically greater cholesterol content than control (approximately 1.7-fold greater than vehicle) (Fig. 5e) but although there was a trend of increased cholesterol levels with Mtb lipid treatment, it did not reach statistical significance for any of the strains (Fig. 5f).

## Exposure to lipids from representative Mtb strains but not *M. canettii* perturb GM1 ganglioside intracellular trafficking

The recycling of glycosphingolipids (GSLs) is perturbed in lysosomal storage diseases, including NPC, resulting in the miss-trafficking of GSLs away from recycling through the Golgi apparatus and instead targets them to the lysosome[20]. This change in lipid trafficking can be visualized by pulse-chase labeling with fluorescently tagged cholera toxin subunit B that binds to GM1 ganglioside at the cell surface[20]. We

used this methodology to assess the extent of altered GM1 trafficking in RAW 264.7 MΦ treated with the different mycobacterial lipids. Confocal microscopy of cells labeled with fluorescently tagged cholera toxin B subunit to monitor trafficking of GM1 ganglioside showed that in wild-type cells GM1 recycles through the Golgi apparatus whereas exposure to U18666A as predicted resulted in GM1 redistribution to the endo-lysosomal compartment (Fig. 6a, b). Similarly, cells treated with lipid extracts from each of the four Mtb strains also had a significant redistribution of the GM1 to the lysosome (Fig. 6a, b), whereas the lysosomal localization of GM1 in MΦ that been exposed to lipids from *M. canettii* was not significantly different from control cells that have a Golgi recycling pattern of staining (Fig. 6a, b).

## GSL species accumulate in RAW 264.7 MΦ incubated with lipid extracts of strains representative of different Mtb lineages but not with *M. canettii* lipid

Impaired lipid transport and catabolism cause the accumulation of multiple different GSLs in NPC cells[21,22]. We therefore extracted GSLs from RAW 264.7 MΦ that had been incubated with cell wall lipid extracts from representative Mtb strains and *M. canettii* and determined their abundance by normal phase HPLC[23]. HPLC profiles representative of U18666A-treated and DMSO are shown in Fig. 7a. As predicted, U18666A significantly increased total GSL content (Fig. 7a, b). Representative HPLC profiles of GSLs from lipid-incubated MΦ extracts and DMSO control are shown in Fig. 7c. Lipid extracts from all four Mtb strains also significantly increased total GSL levels in treated MΦ, but in contrast, total GSLs were significantly lower in cells incubated with *M. canettii* lipids (Fig. 7d). Levels of the most abundant lipid species GM1a, GM1b, and GD1a were significantly increased by lipid extracts from all Mtb strains but in the case of *M. canettii* were either not significantly changed (GM1a) or were significantly reduced (GM1b and GD1a). Strain 281 from Mtb lineage 1 significantly increased eight of the individual lipid species, lineage 2 strain 333 increased five lipid species, and lineage 4 strains 639 and H37Rv increased four and seven distinct lipid species respectively (Fig. 7d and Table 2). *M. canettii* cell wall lipid extract only significantly enhanced the amounts of two minor lipid species (GM2 and GA1), albeit to a relatively small extent (both GSLs were increased by ~1.2-fold by *M. canettii* extract whereas GM2 was increased >2.0-fold and GA1 >1.8-fold by Mtb lipids), did not alter the level of three individual lipids (GA2, Gb3, and GM1a) but significantly reduced the level of four lipids (Lac, GM3, GM1b and GD1a (Table 2), which was not observed with any of the Mtb strains.

## Lipid composition of clinical strains representative of different Mtb lineages

Using a combination of different solvent conditions and thin layer chromatography (TLC), together with matrix-assisted laser

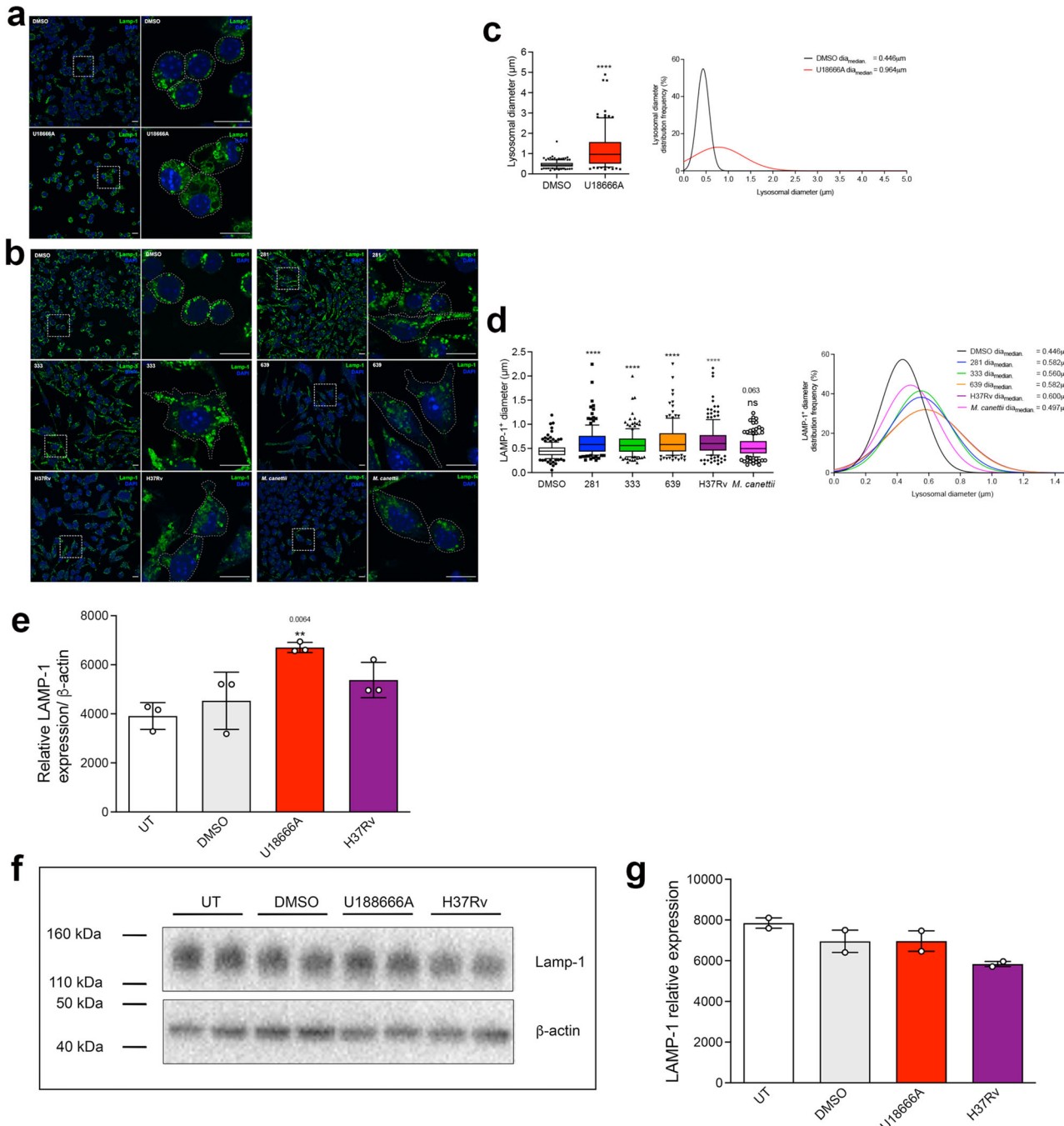

**Fig. 3 | Expansion of the lysosomal compartment in RAW 264.7 MΦ in response to treatment with mycobacterial lipid (100 μg/ml for 48 h) visualized with anti-LAMP-1 staining. a** Confocal microscopy images of RAW 264.7 MΦ treated with vehicle or U18666A and stained with anti-LAMP-1 antibody (green). **b** Images of RAW 264.7 MΦ treated with DMSO or Mtb lipids or *M. canettii* lipid. The right image for each pair is a higher magnification of the region indicated in the left image. Nucleus is stained with DAPI (blue). Scale bar represents 10 μm. **c** Box-and-whisker plot of LAMP-1 positive compartment diameters (μm) and statistical distribution of LAMP-1 positive compartment diameters in vehicle or U18666A treated RAW 264.7 MΦ. Left histogram, data are means ± SEM, *N* = minimum of 100 lysosomes within four random fields. DMSO: min: 0.051 μm, max: 1.190 μm, center: 0.446 μm; U18666A: min: 0.216 μm, max:4.894 μm, center: 0.964 μm. Box contains 90% of all events. Statistical analysis, Mann-Whitney test, two-tailed, ****$p < 0.0001$. Indicated statistical distribution, values are median diameter (μm). **d** Box-and-whisker plot of LAMP-1 positive compartment diameters (μm) and statistical distribution of LAMP-1 positive compartment diameters in DMSO or lipid treated RAW 264.7 MΦ. Data are means ± SEM, *N* = minimum of 100 lysosomes within four random fields. DMSO:

min: 0.051 μm, max: 1.190 μm, center: 0.446 μm; 281: min: 0.255 μm, max: 2.243 μm, center: 0.582 μm; 333: min: 0.204 μm, max: 2.000 μm, center: 0.560 μm; 639: min: 0.253 μm, max: 2.247 μm, center: 0.582 μm; H37Rv: min: 0.180 μm, max: 2.166 μm, center: 0.600 μm; *M. canettii*: min: 0.161 μm, max: 1.232 μm, center: 0.497 μm. Box contains 90% of all events. Statistical analysis, 1-way ANOVA with Kruskal–Wallis test, Kruskal–Wallis statistics: 87.02. ****$p < 0.0001$. MΦ statistical distribution, values indicated are median diameter (μm). Data are representative of two independent experiments. **e** Q-PCR of relative LAMP-1 transcript expression in RAW 264.7 MΦ, normalized to β-actin. Data are mean ± SEM, *N* = 5 replicates per sample. Statistical analysis, *t* test with Welch's correction, two tailed; and Mann–Whitney test, two-tailed. significance value as indicated. **f** Left, Western blot of LAMP-1 and β-actin protein expression in untreated, vehicle, U18666A and H37Rv lipid treated RAW 264.7 MΦ. **g** Quantification of LAMP-1 protein expression relative to β-actin Data are mean ± SEM, *N* = 2 replicates per sample. Statistical analysis, *t* test with Welch's correction No statistically significant differences, two-tailed. Data shown are representative of two independent experiments. Source data are provided as a Source data file.

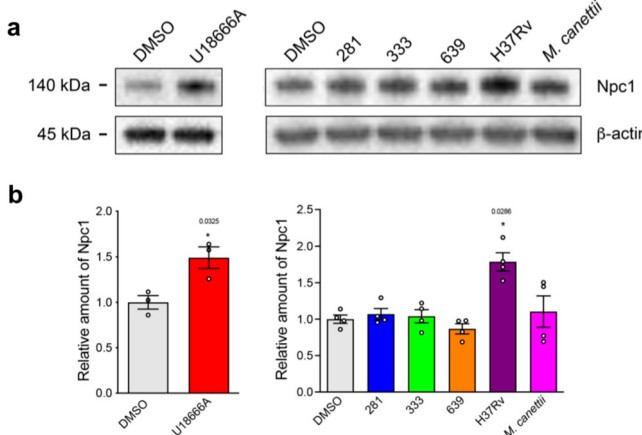

**Fig. 4 | Exposure to lipid extract of strain H37Rv significantly increases the expression of NPC1 in RAW 264.7 MΦ. a** Western blot of protein lysates of RAW 264.7 MΦ that had been incubated with vehicle or mycobacterial lipid extracts for 48 h. Upper band, blot probed for Npc1 and lower band, blot re-probed for β-actin. **b** Quantification of Npc1 protein bands normalized to β-actin, relative to vehicle treatment. Data are mean ± SEM, $N = 2$ or 3. Statistical analysis, $t$ test with Welch's correction, two-tailed. $p$ values as indicated, otherwise not significant. The blot is representative of three independent experiments. Source data are provided as a Source data file.

desorption/ionization-time of flight (MALDI-ToF mass spectrometry (MS)) in the positive and negative ion mode, lipid preparations from each of the Mtb clinical strains were analyzed. As shown in Supplementary Fig. 4a–d, TLC of extractible lipids from the strains produced qualitatively comparable profiles, but no clear differences in composition were apparent. Similarly, in the analysis by MALDI in the negative ion mode, the mass spectra of the mycobacterial strains displayed similar patterns mainly assigned to phosphatidyl-*myo*-inositol-mannosides[24–27] and sulfolipids[28,29] in the *M. tuberculosis* strains (Supplementary Fig. 4e–i). Analysis by MALDI in the positive ion mode revealed a range of lipids, glycolipids, and species-specific lipids (Supplementary Fig. 4e–i).

## Discussion

TB is caused by closely related members of the MTBC family, which have evolved to become facultative intracellular pathogens. Although individual MTBC species have different host preferences, all have adopted a strategy of intracellular survival. The mechanism through which pathogenic mycobacteria survive within host cells following their ingestion is incompletely understood but is important, as it is a major factor contributing to virulence and is a potential clinical intervention point for host-targeted anti-microbial strategies.

In this study, we analyzed the ability of lipid extracts from distinct Mtb strains, other MTBC species, and NTM to inhibit the NPC pathway in host cells. We have found that the ability to increase acidic compartment volume is widespread among the pathogenic mycobacteria examined but was notably absent from *M. canettii*, a smooth bacillus that belongs to a lineage of progenitor species from which it is believed that the MTBC evolved[30–32]. Furthermore, we confirmed that lipid extracts of strains representative of Mtb lineages 1, 2, and 4 but not *M. canettii* induced multiple cellular phenotypes that characterize NPC. We hypothesized that the ability of mycobacteria to inhibit NPC1 will correlate with their ability to establish an intracellular niche and be an important event during the evolution of pathogenicity.

We have previously reported the unexpected phenocopying of the complex cellular phenotypes that occur in the rare lysosomal storage disorder NPC (genetically deficient in *NPC1*) by MΦ infected

with pathogenic mycobacteria[12]. Furthermore, infected cell cultures demonstrated that induction of NPC cellular phenotypes occurred not only in cells harboring mycobacteria but also in bystander non-infected MΦ, indicating that cell wall-derived lipids released from infected cells and endocytosed by neighboring MΦ have identical effects to infection with the live mycobacterium[12].

Whole genome phylogenic studies encompassing the co-evolution of Mtb and humans have identified seven prominent lineages of human-adapted mycobacteria that correlate with spread and geographical distribution[33]. We sampled multiple strains representative of three of the major human geographical lineages[34]: lineage 1, East Africa, Philippines and the rim of Indian Ocean; lineage 2, East Asia; lineage 4, modern MTBC of Europe, America, and Africa and a single strain representative of lineage 3, East African Indian. Lineage 1 is considered representative of ancient human-adapted species, whereas lineages 2 and 4 fall within the classification of modern lineages, based upon the presence or absence of the TbD1 genomic region[35]. Cell wall-derived lipids from 20 strains were chosen to reflect genetic diversity within each lineage[36] and 16 lipid extracts significantly increased lysosomal volume when tested at a concentration of 100 µg/ml, which is consistent with NPC1 inhibition. Of the four strains that were negative, two each were from lineages 1 and 2 and none were from lineage 4, but all were positive when a higher concentration of the lipid extracts was used. The data revealed differential potencies between the strains that could significantly enhance LysoTracker™ staining. It was notable that six strains (119, 173, 293, 367, 440, and the laboratory H37Rv) were capable of increasing acidic compartment volume to a level not significantly different from that induced by the pharmacological NPC1 inhibitor U18666A, which is known to bind to and inhibit NPC1[16] and of these strains, four were from lineage 4 and one each from lineages 1 and 2. Collectively, all lineages induced a significant increase in Lyso-Tracker™ staining. However, we cannot at this point absolutely exclude the possibility that the relative effectiveness between cell wall extracts is a result of the differential recovery of the active lipid species during the extraction process, rather than the inherent bioactive lipid content of each strain. This issue is likely to be resolved when the identity of the active lipid(s) has been determined and fractionation studies are in progress.

One aim of this study was to determine whether the lipid activity in tuberculous mycobacteria that can induce NPC cellular phenotypes is also present in NTM species and may also be involved in promoting their intracellular survival. *M. abscessus* and *M. avium* are opportunistic pathogens that do not cause TB but can be responsible for severe human infections of the respiratory system, skin and mucosa[37–40]. We therefore analyzed lipids prepared from other members of the MTBC (*M. bovis* Bacillus and *M. canettii*) and the atypical or non-tuberculous mycobacteria (NTM), *M. avium* and *M. abscessus*, as well as the environmental non-pathogenic species, *M. smegmatis* (Supplementary Fig. S1). The lack of activity of the lipid extract of the environmental non-pathogenic species *M. smegmatis* was not unexpected as we have previously shown that infection with live *M. smegmatis* did not cause cholesterol accumulation, GSL miss-trafficking, or expansion of lysosomal volume in RAW 264.7 MΦ[12]. Significant expansion of LE/Lys was observed in RAW 264.7 MΦ exposed to lipid extracts of *M. abscessus* and *M. avium*, indicating that the capacity to induce NPC phenotypes is present within the NTB group of mycobacteria. The expanded volume of the acidic compartment following incubation with lipids from specific mycobacteria was confirmed by staining for LysoTracker™ and LAMP-1. Interestingly, we were unable to measure a significant increase in the amount of the LAMP-1 protein, suggesting the pattern of immunoreactivity reflects a redistribution of the protein within enlarged organelles.

The origin and evolutionary history of Mtb and the seven other organisms that comprise the MTBC is of particular interest because of the differences in host specificities and pathogenicity, despite their

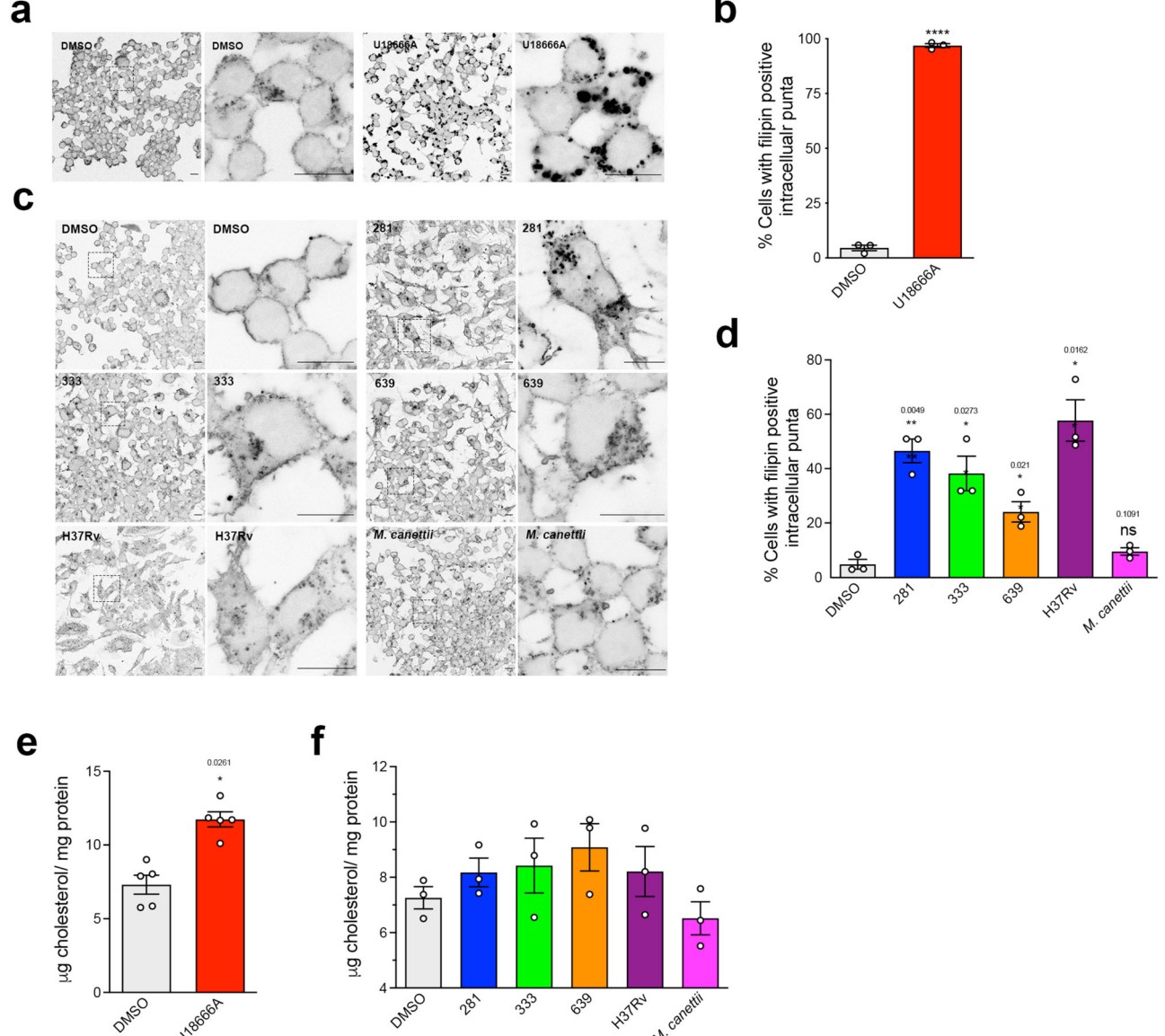

**Fig. 5 | Mtb lipids induce a partial redistribution of cholesterol to intracellular puncta in RAW 264.7 MΦ. a** Confocal microscopy images of vehicle (DMSO) or U18666A treated RAW 264.7 MΦ stained with filipin. For each pair, the right image is a higher magnification of the region outlined in the left image. Scale bar represents 5 μm. **b** Quantitation of the frequency of vehicle or U18666A treated cells with filipin positive intracellular puncta. Data are mean ± SEM, *N* = 3, each representing a minimum of 100 cells in randomly selected fields. Statistical analysis, unpaired *t* test with Welch's correction, two-tailed. ****p < 0.0001. **c** Confocal microscopy images of RAW 264.7 MΦ treated with vehicle (DMSO), Mtb strains or *M. canettii* lipids stained with filipin. For each pair, the right image is a higher magnification of the region outlined in the left image. Scale bar represents 5 μm. **d** Quantitation of the frequency of DMSO, Mtb lipids, or *M. canettii* lipid treated cells with filipin positive intracellular puncta. Data are mean ± SEM, *N* = 3, each representing a minimum of 100 cells in randomly selected fields. Statistical analysis, unpaired *t* test with Welch's correction, two-tailed. *p* values as indicated. **e** Amplex red quantitation of cholesterol content of RAW 264.7 MΦ treated with vehicle (DMSO) or U18666A. Data are mean ± SEM, *N* = 3 replicates per sample. Statistical analysis, unpaired *t* test with Welch's correction, two-tailed. Significance value as indicated. **f** Amplex red quantitation of cholesterol content of RAW 264.7 MΦ treated with vehicle (DMSO) or mycobacterial lipids. Data are mean ± SEM, *N* = 5. Statistical analysis, unpaired *t* test with Welch's correction, two-tailed. No statistically significant differences. Data are representative of three independent experiments. Source data are provided as a Source data file.

remarkable level of sequence similarity at the nucleotide level[30,41]. The seven human-adapted lineages are not considered monophyletic. The consensus from multiple genetic studies is that members of the MTBC emerged from progenitor species that likely resembled the smooth tubercule *M. canettii*, which itself diverged prior to the MTBC[42]. It was therefore of significant interest that incubation of RAW 264.7 MΦ with the lipid extract from *M. canettii* did not induce any NPC cellular phenotypes, suggesting that the bioactive lipid(s) is either lacking or is of very low abundance in this mycobacterium. Biochemical analysis of mycobacterial cell wall lipids has revealed a discontinuous

distribution, consistent with the evolutionary emergence of Mtb from an environmental species resembling *Mycobacterium kansasii* via intermediate smooth tubercle bacilli[43]. The diversity of *M. canettii* lipids includes some that are unique, such as the phenolic glycolipids phenolphthiocerol dimycocerosates that have been lost during the subsequent emergence of MTBC[44,45]. Molecular and phenotypic studies of the smooth tubercle *M. canettii* have provided evidence for delineating the evolution of Mtb, elucidating the underlying mechanisms that were responsible and the development of pathogenicity/intracellular survival and virulence[46,47]. *M. canettii* strains display

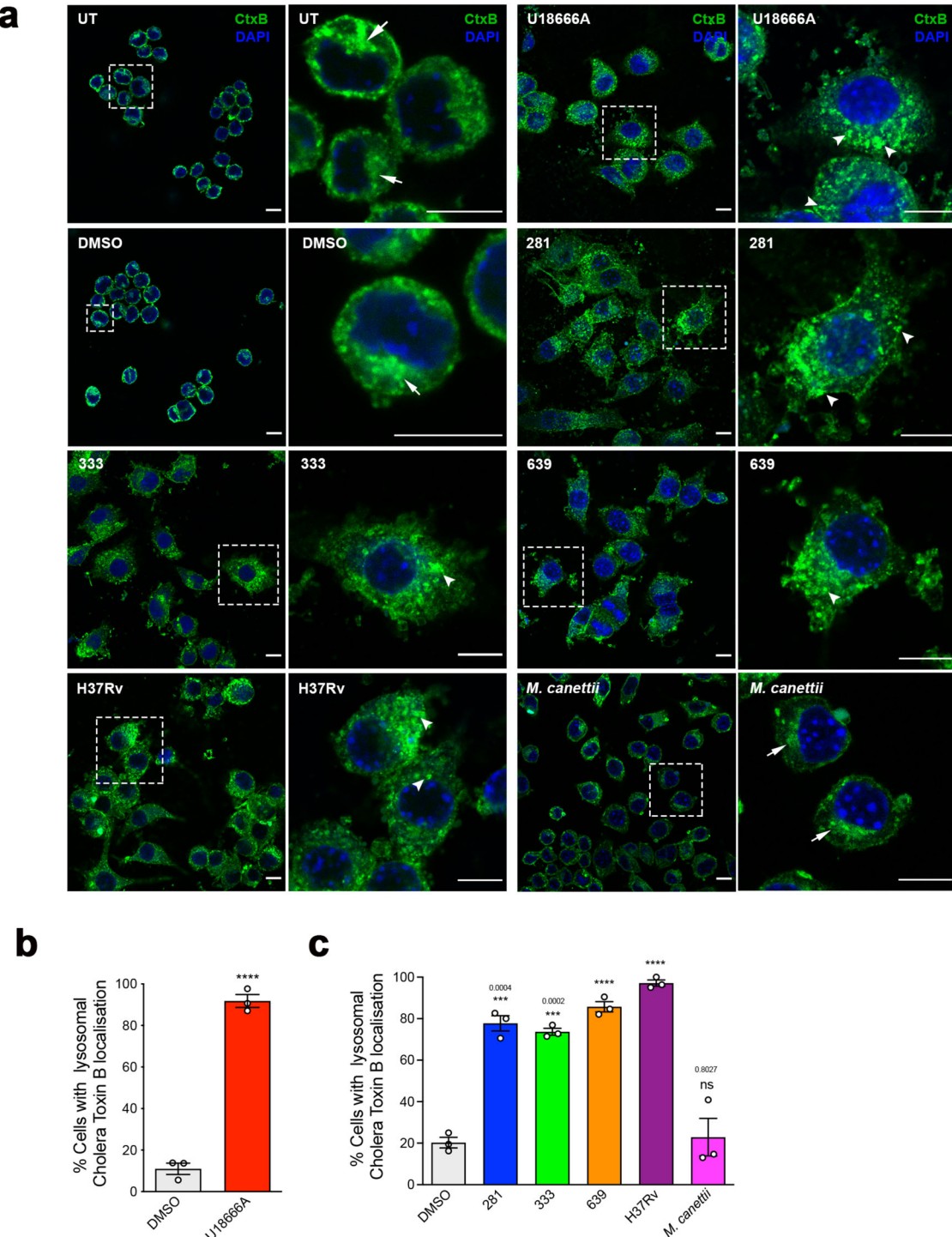

**Fig. 6 | Mtb lipids affect GM1 trafficking in RAW 264.7 MΦ. a** Confocal micro-scopy images of RAW 264.7 MΦ treated with vehicle (DMSO), U18666A, or mycobacterial lipids and pulse-chased with cholera toxin B subunit (green). Nucleus is stained with DAPI (blue). For each pair, right image is a higher magni-fication of the region outlined in the left image. Untreated, vehicle and *M. canettii* lipid treated cells show a predominantly Golgi pattern of staining, indicated by the white arrows. U18666A and lipids from Mtb strains induce a punctate pattern, consistent with lysosomal distribution (white arrowheads). Scale bar represents 10 μm. **b** Percentage of vehicle and U18666A treated cells with lysosomal cholera toxin B localization. Data are mean ± SEM, $N = 3$, each representing a minimum of 100 cells in randomly selected fields. Statistical analysis, unpaired *t* test with Welch's correction, two-tailed. ****$p < 0.0001$. **c** Percentage of the vehicle and mycobacterial lipid-treated cells with lysosomal cholera toxin B localization. Data are mean ± SEM, $N = 3$, each representing a minimum of 100 cells in randomly selected fields per sample. Statistical analysis, unpaired *t* test with Welch's cor-rection, two-tailed. ****$p < 0.0001$, other *p* values as indicated. Data are repre-sentative of two independent experiments. Source data are provided as a Source data file.

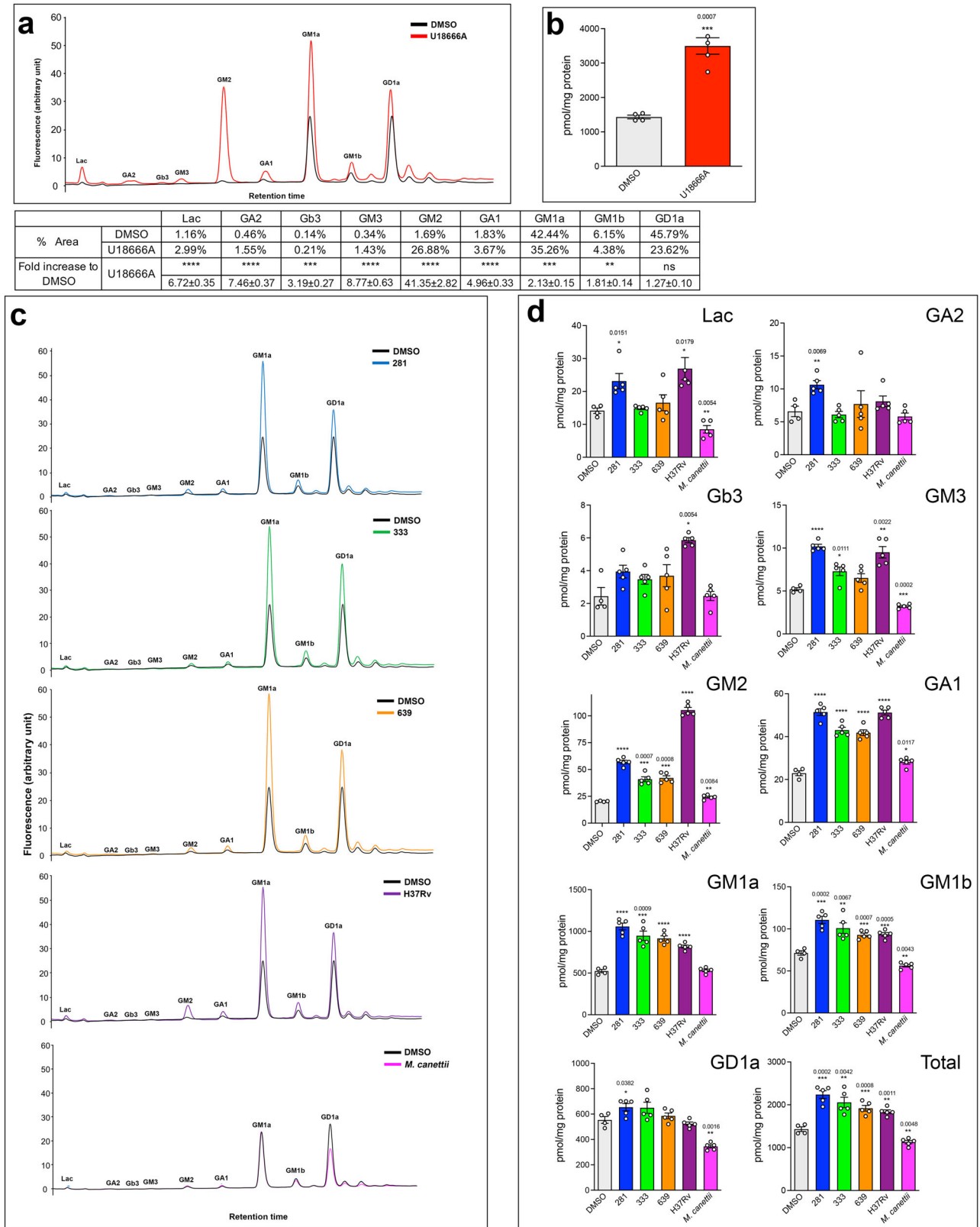

**Fig. 7 | Mtb lipids, but not *M. canettii* lipids cause GSL accumulation in RAW 264.7 MΦ. a** Representative HPLC trace of GSLs from vehicle (DMSO) and U18666A treated cells are overlaid with individual GSL peaks annotated. The table summarizes the percent area of each peak and the fold increase induced by U18666A. **b** Quantitation of total GSLs in DMSO or U18666A treated cells. Data are mean ± SEM, *N* = replicates per sample. Statistical analysis, unpaired *t* test with Welch's correction, two-tailed. ***p < 0.0007. **c** Overlays of representative HPLC

traces of extracts of RAW 264.7 cells treated with vehicle (DMSO) and lipid extracts of Mtb strains or *M. canettii*. **d** Quantitation of individual GSL species and total GSLs in DMSO or mycobacterial lipid treated cells. Data are mean ± SEM, *N* = 4–5 replicates per sample. Statistical analysis, unpaired *t* test with Welch's correction, two-tailed ****p < 0.0001, other *p* values as indicated. Source data are provided as a Source data file.

**Table 2 | Summary of relative accumulation of total and specific GSL species in RAW 264.7 MΦ treated with U18666A or lipid extracts from different Mtb clinical strains and *M. canettii***

| | U18666A | 281 | 333 | 639 | H37Rv | *M. canettii* |
|---|---|---|---|---|---|---|
| Lac | **** <br> 6.72 ± 0.35 | *0.0151 <br> 1.64 ± 0.16 | ns <br> 1.05 ± 0.03 | ns <br> 1.17 ± 0.17 | *0.0179 <br> 1.90 ± 0.24 | **0.0054 <br> 0.60 ± 0.08 |
| GA2 | **** <br> 7.46 ± 0.37 | **0.0069 <br> 1.62 ± 0.10 | ns <br> 0.93 ± 0.07 | ns <br> 1.17 ± 0.31 | ns <br> 1.23 ± 0.12 | ns <br> 0.88 ± 0.08 |
| Gb3 | ***0.004 <br> 3.19 ± 0.27 | ns <br> 1.62 ± 0.16 | ns <br> 1.42 ± 0.12 | ns <br> 1.51 ± 0.28 | *0.0054 <br> 2.40 ± 0.07 | ns <br> 1.01 ± 0.12 |
| GM3 | **** <br> 8.77 ± 0.63 | **** <br> 1.96 ± 0.05 | *0.0111 <br> 1.40 ± 0.10 | ns <br> 1.52 ± 0.10 | **0.0022 <br> 1.83 ± 0.13 | ***0.002 <br> 0.62 ± 0.02 |
| GM2 | **** <br> 41.35 ± 2.82 | **** <br> 2.82 ± 0.08 | ***0.0007 <br> 2.02 ± 0.11 | ***0.008 <br> 2.08 ± 0.10 | **** <br> 5.21 ± 0.12 | **0.0084 <br> 1.20 ± 0.04 |
| GA1 | **** <br> 4.96 ± 0.33 | **** <br> 2.24 ± 0.07 | **** <br> 1.88 ± 0.06 | **** <br> 1.82 ± 0.06 | **** <br> 2.23 ± 0.05 | *0.0117 <br> 1.23 ± 0.04 |
| GM1a | ***0.0006 <br> 2.13 ± 0.15 | **** <br> 2.02 ± 0.08 | ***0.0009 <br> 1.81 ± 0.11 | **** <br> 1.75 ± 0.06 | **** <br> 1.56 ± 0.03 | ns <br> 1.02 ± 0.03 |
| GM1b | **0.0071 <br> 1.81 ± 0.14 | ***0.0002 <br> 1.55 ± 0.06 | **0.0067 <br> 1.41 ± 0.09 | ***0.0007 <br> 1.30 ± 0.03 | ***0.0005 <br> 1.31 ± 0.03 | **0.0043 <br> 0.79 ± 0.02 |
| GD1a | ns <br> 1.27 ± 0.10 | *0.0382 <br> 1.18 ± 0.05 | ns <br> 1.17 ± 0.08 | ns <br> 1.06 ± 0.04 | ns <br> 0.95 ± 0.03 | **0.0016 <br> 0.62 ± 0.02 |
| Total GSL | ***0.0003 <br> 2.44 ± 0.17 | ***0.0002 <br> 1.56 ± 0.06 | **0.0042 <br> 1.44 ± 0.09 | ***0.0008 <br> 1.34 ± 0.05 | **0.0011 <br> 1.28 ± 0.03 | **0.0048 <br> 0.79 ± 0.03 |

Summary of relative abundance of total GSL content and individual lipid species of RAW 264.7 MΦ treated with lipid extract from different *Mtb* strains or *M. canettii*. Data represent mean ± SEM fold change in lipid abundance relative to DMSO-treated control cells. Statistical analysis, 2-way ANOVA, $N$ = minimum of 4-5 samples per treatment. ****$p$ < 0.0001, other $p$ values as indicated, *ns* not significant. Data are representative of two independent experiments. Source data are provided as a Source data file.

significant genome variation[48,49], which sets them apart from other MTBC. Clinical infection with *M. canettii* is rare and geographically restricted, while variants show differential virulence, all of which indicate that it is far less successful at causing human disease in comparison to Mtb[46,50]. It is therefore possible that the failure of *M. canettii* lipids to inhibit NPC1 indicates the use of an alternative mechanism to facilitate infection and intracellular survival that is overtly less effective than the NPC1 inhibition strategy exploited by Mtb and other mycobacterial species such as *M. bovis Bacillus* and *M. avium*.

The expansion of the acidic compartment, both in terms of lysosome number and size of individual lysosomes following lipid treatment was also confirmed by confocal microscopy using LysoTracker™ and anti-LAMP-1 staining. Although microscopy images of cells stained for the lysosomal protein, LAMP-1 were suggestive of greater expression, analyses of both mRNA levels and protein expression did not confirm an increase, implying that over the time course that was used, it was more likely that there was a re-distribution of the protein in the expanded lysosomes.

Previously, we have reported that infection of RAW 264.7 MΦ with BCG causes a significant upregulation of NPC1 protein[12]. When we investigated the capacity of mycobacterial lipids to induce the same effect, only the lipid extract of H37Rv significantly enhanced NPC1 expression. Currently, we are unable to fully explain the differences between these data, but one possibility is that infection with live mycobacteria achieves greater inhibition of NPC1 activity, and as a consequence protein abundance is increased in an attempt to compensate for the loss of NPC1 function.

We also investigated the capacity of mycobacterial lipid extracts to cause lysosomal accumulation of cholesterol in RAW 264.7 MΦ, which is a prominent cellular NPC phenotype[14]. The association between cholesterol and Mtb infection has been recognized in multiple studies. It is essential for the entry of mycobacteria into MΦ[51], its utilization is required for survival[19] and it is important for maintaining dormancy and enabling reactivation[52]. Cholesterol is also used as a critical energy source illustrated by the presence within the mycobacterial genome of genes encoding transport proteins that mediate

its import and enzymes necessary for its metabolism[53,54]. We detected between a five and ten-fold increase in the frequency of cells that had intracellular filipin-positive punta with Mtb lipid treatment. To our knowledge, the comparative ability of different Mtb strains and members of MTBC to manipulate host cell cholesterol has not been explored previously.

Consistent with inhibition of NPC1, lipid extracts from the mycobacteria that affected cholesterol accumulation also caused miss-localization of GM1 ganglioside, which did not occur when RAW 264.7 cells were treated with lipids from *M. canettii*. At this point, it is not obvious if pathogenic mycobacteria directly benefit from biochemical changes in the intracellular distribution of host GSLs that result from blocking NPC1-dependent activities and this merits further investigation.

The mycobacterial envelope is architecturally and biochemically complex, is composed of multiple carbohydrates and lipid species[55,56], and functions as a permeability barrier[56,57]. Intercalated within the lipid environment of the cell wall are extractable free lipid species that can influence the balance between pathogen and host by affecting pathogenesis and shaping immune responses[55,58,59]. In light of the findings that lipid extracts from strains representative of the three Mtb lineages increased the relative LE/Lys volume to differing extents, we performed lipidomic analyses of sixteen clinical isolates, plus the laboratory strain, H37Rv, to explore lipid diversity and the possibility of correlation between bioactivity and lipid composition. We focused on extractable lipids. As seen on the TLCs, qualitatively no differences were observed, whereas quantitatively, there were some differences in the levels/ratios of the respective lipids within and across lineages. MALDI analysis of the mycobacterial strains confirmed these differences: mass spectra in the negative ion mode displayed similar patterns, mainly assigned to phosphatidyl-*myo*-inositol-mannosides[24–27] and sulfolipids[28,29]. However, mass spectra in the positive ion mode revealed a range of lipid, glycolipids, and species-specific lipids, which is in accordance with the published literature[24,60,61]. These differences could be a factor influencing the phenotypes observed. In addition, it is known that one of the major surface exposed lipids, lipooligo-saccharides (LOS), are absent in Mtb as opposed to *M. canettii* due to

the mutation within the *pks5* locus abrogating synthesis[42]. LOS could therefore be an ideal candidate to explain the suppression of the phenotypes in *M. canettii* compared to Mtb, masking more potent surface-exposed lipids in the strains used in this study. The lack of activity of the *M. smegmatis* lipid extract could perhaps be explained by an inadequate quantity of specific lipid(s) or the presence of a significant level of exposed glycopeptidolipids[62,63]. It would be of interest in future studies to analyze the lipid extract of *M. canettii* rough morphotype variants I and K, which are deficient for LOS, and test for its capacity to induce NPC1 inhibition.

More specifically, based on the evolution of MTBC due to genetic deletions or rearrangement of regions of the genome that contains enzymes involved in the synthesis of surface exposed lipids and gly-colipids, some lipids found in *M. canettii* cannot be produced by Mtb H37Rv[35,46,55,64,65]. Indeed, through evolution, Mtb has lost the ability to produce lipooligosaccharides because of pks5 locus recombination[42,66]. With respect to the biochemical differences between Mtb H37Rv and NTM, diglycosylated glycopeptidolipids and triglycosylated glyco-peptidolipids make up more than 70% of the surface-exposed myco-bacterial lipids in *M. abscessus* are not found in Mtb H37Rv[67–71]. In addition, mycolic acids, such as ones bound to trehalose in TMM and TDM, and particularly their decorations can differ between myco-bacterial species[72,73]. Indeed, mycolic acids display high structural diversity with variations observed in chain length (60 to 90 carbon atoms), the extent of unsaturation, and chemical groups, such as ketones and methoxys[73]. This high level of diversity provides mycolic acids with species-specific characteristics[74]. In addition to mycolic acids, other lipids are also specific to particular species of myco-bacteria. For example, sulfolipid 1 and polyacyltrehalose are found exclusively in *M. tuberculosis*, while trehalose polyphleate is present in NTM species[28,75,76]. Layre et al.[77] using an HPLC–MS-based lipidomics platform screened for lipids that were present in *M. tuberculosis* but not in *M. bovis* BCG and identified 1-tuberculosinyladenosine (1-TbAd), an amphipathic diterpene nucleoside. However, its presence or absence in NTM has not as yet been reported.

In summary, the findings presented here indicate that the lipid-dependent capacity to trigger specific host cell phenotypes indicative of NPC1 inhibition is broadly distributed among pathogenic myco-bacteria species and Mtb clinical strains. The lack of activity in extracts of *M. canettii* smooth morphotype is consistent with the hypothesis that the evolution of the lipid(s) that is responsible for NPC1 inhibition occurred after the divergence of MTBC from the ancestral *M. canettii*-like species. Future identification and characterization of the lipid will facilitate a greater mechanistic understanding of how pathogenic strains establish infection and maintain intracellular survival.

## Methods

### Mammalian cell culture
The murine macrophage (MΦ) cell line RAW 264.7 was obtained from the ATCC and maintained in RPMI 1640 containing 10% (v/v) fetal bovine serum, 1% L-glutamine, and 1% penicillin–streptomycin (all Sigma-Aldrich) at 37 °C in 5% CO2. Cells were passaged routinely to maintain viability (>90%).

### Bacterial strains and culture conditions
*M. tuberculosis* strains were cultured in Sauton's medium as cell surface pellicle and prepared at Imperial College, University of London: Mtb clinical strains consisted of 16 clinical strains of Mtb collected in Vietnam[36]: 4 Indo-Oceanic (Lineages 1 or named Group I), 6 Beijing (Lineage 2 named Group II) and 6 Euro-American (Lineage 4 or named Group III) to reflect the genotypic diversity found within each lineage in the study Group I (232, 281, 346, 372); Group II (119, 212, 333, 345, 374, 639); Group IIII (173, 293, 318, 367, 440, 639) and Mtb H37Rv: non-tuberculous mycobacteria: *Mycobacterium abscessus* (ATCC19977) and *Mycobacterium avium* (ATCC 25291) and the environmental species

*Mycobacterium smegmatis* mc²155 (ATCC 700084). The following reagents were obtained through BEI Resources, NIAID, NIH: Mtb, Strain, CDC1551, total lipids, NR-14838, Mtb, Strain East African Indian 91_0079, NR-44098, Mtb Strain H37Rv, total lipids, NR-14837, *Mycobacterium canettii* total lipids, NR-40334 and *Mycobacterium bovis*, Strain AF 2122/97 (ATCC BAA-935), total lipids, NR-44100, Mtb Strain H37Rv, Purified Mycolylarabinogalactan-Peptidoglycan, NR-14851, Mtb Strain H37Rv, Purified Lipomannan, NR-14850, Mtb Strain H37Rv, Purified Lipoarabinomannan, NR-14848 and Mtb Strain H37Rv, Purified Arabinogalactan, NR-14852.

### Lipid solubilization and addition to RAW 264.7 MΦ
Dried lipid extracts were re-constituted at a concentration of 20 mg/ml in DMSO by sonicating for 2 min at 60 °C in an ultrasonic water bath. DMSO stocks were diluted to the final concentration in pre-warmed culture medium, sonicated for 2 min at 60 °C, allowed to cool, and immediately added to cells. RAW 264.7 cells were plated at a density of $5 \times 10^4$ cells/well in 96-well culture plates in a complete medium and allowed to adhere overnight. Lipid extracts, 2 µg/ml U18666A (Merck) or vehicle (DMSO only) were added and plates were incubated at 37 °C, 5% CO2 for 48 h before analysis.

### LysoTracker™ staining: FACS analysis
LysoTracker™ staining was performed as described previously[15]. In brief, MΦ were harvested, washed twice with PBS, and stained with 200 nM LysoTracker™ Green DND-26 (ThermoFisher) for 10 min in the dark. Cells were washed with PBS and re-suspended in buffer con-taining 5 µg/ml Propidium iodide (Sigma) to allow for the exclusion of dead cells and immediately analyzed on a BD FACS-Canto II (Beckton Dickinson). A minimum of 10,000 events were collected for each sample using DIVA software v*.0.1 (Beckton Dickinson) and relative fluorescence values calculated using the FlowJo software (Version 10, FlowJo, LLC). The experimental design of each independent investi-gation ensured that all samples could be analyzed within 45 min of LysoTracker™ staining to ensure consistency of fluorescent signal[15].

### LysoTracker™ staining: confocal microscopy
LysoTracker™ live cell imaging was performed using Cellview cell culture slides (Greiner). After vehicle, U18666A or lipid treatments, cells were washed with PBS and stained with 200 nM LysoTracker™ Red DND-99 (ThermoFisher) for 10 min in the dark. Cells were washed with PBS, stained with 200 nM Hoechst 33342 nuclear stain (Thermo-Fisher) for 5 min, and washed again. Cells were kept in PBS and imaged within 1 h of staining on a Leica-SP8 confocal microscope.

### Western blotting
Cells were harvested and lysed with RIPA buffer (ThermoFisher) containing cOmplete™ EDTA-free protease inhibitor cocktail (Merck), centrifuged and the supernatant collected. The protein concentration of the supernatant was determined by BCA assay (Sigma). Volumes of cell lysate containing 20 µg protein were mixed with SDS Blue loading buffer (BioLabs) and dithiothreitol (DTT, BioLabs), and heated at 50 °C for 10 min. Samples were loaded onto NuPAGE™ 4–12% Bis-Tris Gels (ThermoFisher) and run in NuPAGE™ MOPS SDS running buffer (ThermoFisher). Novex Sharp Pre-stained protein standard (ThermoFisher) was used to indicate the extent of protein migration. Gels were transferred onto Immuno-Blot® PVDF membrane (Bio-Rad) using BIORAD transblot turbo transfer system (Bio-Rad). Membranes were blocked with 5% skimmed milk in PBS with 0.1% Tween 20 (Sigma) for 1 h at room temperature and then incubated with primary antibody (monoclonal rabbit anti-LAMP-1 antibody EPR21026, 1:1000, Abcam ab208943, polyclonal rabbit anti-NPC1 antibody NB400–148, 1:1000, Novus) diluted in PBS con-taining 2.5% skimmed milk with 0.1% Tween 20 and sodium azide overnight at 4 °C. Membranes were washed with PBS containing 0.1%

Tween 20 three times for 20 min each and then incubated with HRP-conjugated anti-rabbit secondary antibody (1:5000, Sigma A0545) diluted in PBS containing 2.5% skimmed milk and 0.1% Tween 20 for 1 h at room temperature. Membranes were washed as before and then developed with SuperSignal™ West Femto substrate (Thermo-Fisher) or Pierce™ ECL western blotting substrate (ThermoFisher). Membranes were re-probed with peroxidase-conjugated anti-β-actin antibody AC-15, 1:10,000 (Sigma A3845) to evaluate equivalent protein loading. Images were obtained using a ChemiDoc XRS system (BioRad) and processed and analyzed with ImageLab software (BioRad) with BioRad Universal Hood.

## Immunofluorescence staining

After lipid treatment, cells were washed with PBS and fixed with 4% paraformaldehyde (PFA) for 15 min at room temperature. Cells were then permeabilized with 0.1% saponin in PBS for 15 min at room temperature and washed with PBS and blocked with 5% goat serum in PBS for 45 min at room temperature. Cells were incubated with a primary anti-LAMP-1 antibody with 10% goat serum and 0.1% saponin in PBS (rat monoclonal, clone 1D4B 1:500, Abcam ab25245) overnight at 4 °C and then with anti-rat IgG secondary antibody (1:1000, ThermoFisher A21208) for 45 min at room temperature. Cells were stained with DAPI (4′,6-diamidino-2-phenylindole, ThermoFisher) to visualize nuclei and mounted with Prolong Gold Antifade (ThermoFisher). Imaging was carried out on a Leica-SP8 confocal microscope.

## Filipin Staining for free cholesterol

For labeling of free cholesterol, cells were fixed with 4% PFA for 15 min at room temperature and washed with PBS, then incubated with 1.5 mg/ml glycine (Sigma) in PBS for 10 min to quench aldehyde groups. Cells were then stained with a Filipin working solution composed of 0.05 mg/ml filipin complex from *Streptomyces filipinensis* (Sigma), 10% FBS (Sigma), and 0.2% Triton x100 (Sigma) in PBS for 2 h at room temperature. Cells were washed with PBS and stained with propidium iodide (PI, 1:3000, Sigma) to visualize nuclei and mounted with Prolong Gold Antifade (ThermoFisher). Imaging was carried out on a Leica-SP8 confocal microscope.

## Amplex red measurement of cholesterol

Cells were lysed in glass tubes in 0.5 ml of ddH$_2$O and freeze/thawed three times. 2 ml chloroform: methanol (2:1, v/v, Sigma) was added, samples homogenized for 5 min and then mixed at room temperature for 2 h on a roller. 0.4 ml methanol was then added and samples were centrifuged for 10 min at $1500 \times g$ and supernatant recovered. 0.8 ml chloroform and 0.73 ml ddH$_2$O were added to a final composition of CH$_3$Cl: MeOH: ddH$_2$O (8:4:3, v/v/v). Lysates were mixed and centrifuged, the upper phase removed, and the lower phase washed with CH$_3$Cl: MeOH: ddH$_2$O (3:48:47, v/v/v) 3 times and then dried under N$_2$. Total cholesterol in the samples was determined using Amplex Red Cholesterol Assay Kit (Sigma) according to the manufacturer's protocol.

## Cholera toxin subunit B (CtxB) staining

After treatment with lipid or drug, cells were washed with PBS and incubated with 0.5ug/ml CtxB in PBS for 30 min on ice. Cells were washed with PBS and labeled GM1 was chased for 1 h with fresh media. Cells were washed with PBS containing 1% normal goat serum and fixed with 4% PFA for 15 min at room temperature. Nuclei were stained with DAPI and mounted with Prolong Gold Antifade (ThermoFisher). Imaging was carried out on a Leica-SP8 confocal microscope.

## Analysis and quantification of GSLs by normal phase high-performance liquid chromatography (NP-HPLC)

GSLs were analyzed essentially as described previously[23]. Aqueous extracts of frozen pellets of control and lipid-treated RAW 264.7 MΦ

were prepared by three cycles of freeze/thawing. Protein concentration was determined by BCA assay (Sigma). Lipids were extracted from aqueous cell homogenates (approximately 0.2 mg in 0.2 ml) with chloroform and methanol overnight at 4 °C. GSLs were further purified using solid-phase C18 columns (Telios, Kinesis) and eluted. Fractions were dried down under a stream of nitrogen at 42 °C and treated with recombinant ceramide glycanase (rEGCase; GenScript) to release ceramide-linked glucosylceramide-derived GSL oligosaccharides. The liberated free glycans were fluorescently labeled at 80 °C for 60 min with anthranilic acid (2AA) and sodium cyanoborohydride. Purification of labeled glycans and removal of excess 2AA was achieved by passing reaction mixes over DPA-6S SPE columns (Supelco). 2AA-labeled oligosaccharides were separated and quantified by normal-phase HPLC as described by the authors in ref. [23]. The NP-HPLC equipment consisted of a Waters Alliance 2695 separation module and an in-line Waters 2475 multi λ-fluorescence detector set at Ex λ 1360 nm and Em λ 1425 nm. The solid phase was a $4.6 \times 250$ mm TSK gel-Amide 80 column (Anachem). Glucose unit values (GUs) were determined using a 2AA-labeled glucose homopolymer ladder (Ludger). To calculate molar quantities from integrated peaks in the chromatogram, 2.5 pmol 2AA-labeled chitotriose (Ludger) was injected as a calibration standard with each sample set.

## qPCR

RNA from RAW 264.7 MΦ was isolated using Monarch total RNA MiniPrep kit (BioLabs) according to the manufacturer's protocol. cDNA was synthesized using iScript cDNA synthesis kit (BIO-RAD). Reaction mixes containing cDNA template, PowerUp SYBER Green (ThermoFisher), and primers specific for LAMP-1 gene expression level were run on a CFX96 Real-Time PCR system (BIO-RAD). Assays specific for relative expression of murine β-actin was used to normalize for input.

## TLC of extractible lipids and MALDI lipid fingerprinting

Heat-killed mycobacteria were first washed 3 times with PBS. The pellets were then submitted to CHCl$_3$/MeOH 1:2 (v/v) extraction for 12 h at room temperature followed by one CHCl$_3$/MeOH 1:1 (v/v) extraction and one CHCl$_3$/MeOH 2:1 (v/v) extraction for 3 h at room temperature. Pooled extracts were concentrated and evaporated to dryness. Total lipids extracted were normalized to the dried weight of the lipid extract. The dried lipid extracts were resuspended in CHCl$_3$ at a final concentration of 20 mg/ml. Five microliters, equivalent to 100 µg, were loaded on TLC. TLCs were run in four different solvent systems: Petroleum Ether/Diethyl Ether 95:5 (v/v); CHCl$_3$/MeOH 9:1 (v/v); CHCl$_3$/MeOH 8:2 (v/v) and CHCl$_3$/MeOH/H$_2$O 60:25:4 (v/v/v). Then, TLCs were sprayed with a solution of 5% phosphomolybdic acid in 100% ethanol and heated at 200 °C for 2 min in order to reveal the lipids. All experiments were performed in triplicate for statistical confidence.

For lipid fingerprinting, 0.5 µg of total lipid extract was loaded onto the target and immediately overlaid with 0.5 µl of a 2,5-dihydroxybenzoic acid (DHB) matrix used at a final concentration of 10 mg/ml in chloroform/methanol (CHCl$_3$/MeOH) 90:10 v/v. Bacterial total lipid extracts and matrix were mixed directly on the target by pipetting and the mix was dried gently under a stream of air. MALDI-TOF MS analysis was performed on a 4800 Proteomics Analyzer (Applied Biosystems) using the reflectron mode. Samples were analyzed by operating at 20 kV in the positive and negative ion mode using an extraction delay time set at 20 ns. Data were obtained using Data Explorer version 4.9 (build 115) Applied Biosystems.

## Image acquisition and processing

Confocal images were acquired using Leica TPC SP8 confocal microscope and LAS X software. Images for LysoTracker™ Red DND-99, Lamp-1, and GM1 trafficking were captured using a ×60

oil-immersion (Type F immersion oil, Leica) objective with the 488 nm laser for excitation and 690 nm laser for emission (Lyso-Tracker™ Red setting for LysoTracker™ Red DND-99 LFITC for Lamp-1; Alexa 488 for GM1). Lysosomal GM1 was identified as fluorescent Cholera Toxin subunit B positive vesicles (puncta) in the endo-lysosome, whereas Golgi distributed GM1 was identified by a peri-nuclear pattern of CtxB staining. Counts were made from a minimum of 100 cells across three random fields. Filipin images for cholesterol distribution were obtained using x40 oil-immersion objective with 405 nm laser for excitation and 727 nm laser for emission with 0.75% UV. Data for intracellular cholesterol accumulation were determined as the percentage of cells with lysosome distributed puncta. For each sample, at least 3 random fields were examined, and counts were made of a minimum of 100 cells in each. Nuclear staining was captured using 405 nm laser for excitation, 480 nm laser for emission for DAPI and Hoechst 33342 staining, or 552 nm laser for excitation, 786 nm laser for emission for PI staining. Lysosomal diameter was determined by analysis of LysoTracker™ Red DND-99 and LAMP-1 staining of treated cells. Punctate lysosomes in RAW 264.7 MΦ were defined from 0 to 0.6μm, and enlarged lysosomes were defined by having diameters from 0.6 to 5 μm. A minimum of 100 lysosomes were measured in at least three random fields and analyzed. Confocal images were analyzed using ImageJ Fiji (NIH, USA).

### Statistical analysis

Normality of data was determined before performing statistical analyses. Analysis and statistical significance of all data sets were determined using GraphPad Prism 9.

### Reporting summary

Further information on research design is available in the Nature Research Reporting Summary linked to this article.

## Data availability

Source data are provided with this paper.

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

## Acknowledgements

We thank Jiayun Zhu for her preliminary data on mycobacterial lipid analysis. This study was funded by an Investigator in Science Award from The Wellcome Trust to F.M.P. (202834/Z/16/Z).

## Author contributions

Conceptualization F.M.P., G.L.-M., N.P., B.D.R.; methodology Y.W., D.S., Y.L.; validation Y.W., D.S., Y.L., N.P.; investigation Y.W., D.S., Y.L.; data curation Y.W., D.S., Y.L., N.K.; visualization Y.W., D.S., N.P., Y.L., G.L.-M.; writing—original draft N.P.; writing—review and editing Y.W., D.S., Y.L., N.K., B.D.R., N.P., G.L.-M., and F.M.P.; supervision F.M.P., G.L.-M., and N.P.; funding acquisition F.M.P. and G.L.-M.

## Competing interests

F.M.P. is a consultant to and co-founder of IntraBio. The other authors declare no competing interests.
