## [Peer Review File · Nature Communications]

REVIEWER COMMENTS

Reviewer #1 (Remarks to the Author):

Weng and colleagues report on the role of early evolution of lipids capable of inhibiting NPC1 and shows divergence from *Mycobacterium canettii*-related mycobacteria. The study builds on a previous study (Fineran et al. 2016) which reported that lipids shed by persistent mycobacteria inhibit NPC1 which causes drop in lysosomal calcium levels stalling phagosome-lysosome fusion and thereby leads to mycobacterial persistence. In the current study, the authors demonstrate that unlike common ancestor, *Mycobacterium canettii*, the cell wall lipid extracted from diverse pathogenic mycobacterial species induce NPC-deficient macrophage like phenotype and have correlated the inhibition to NPC1 with virulence and ability of mycobacteria to persist.

The word “persist” can only be used in the context of “long-term disease persistence” or “antibiotic persistence” (Balaban et al., 2019, *Nature Reviews Microbiology* 17, 441-448) neither of which has been demonstrated to be modulated by NPC1 inhibition. Hence use of the word “survive” sounds more appropriate.

The evidence that Mtb lipids directly modulate NPC1 activity is missing. The work, however, seems preliminary and contains many results that are poorly related. The reader is finally left with a very interesting hypothesis that is not sufficiently explored mechanistically and needs substantial improvement which could be strengthened by additional experiments using cell line and in-vivo and ex-vivo experiments using NPC1-/- knock-out mice and different mycobacterial strains.

Major Comments:

1. Figure 1: Not sure why HupB has been used to determine phylogenetic relatedness? In my opinion, housekeeping genes or genes involved in lipid metabolism could have been used instead.
2. Figure 2: The fact that not all the strains (2/5) for L1 and (2/6) for L2 adhere to the said hypothesis, attributed to the potency and variation in lipid components between strains, can only be reconciled by identifying the active component(s) directly modulating NPC1 function.
3. Figure 3: Quantification is missing. The authors should show a larger field of interest. The conclusion that “..LAMP1 immunoreactivity was higher in lipid-treated cells...” is debatable and I do not see any difference in the immunoreactivity to LAMP1 antibodies in any of the panels except for the U186660 treated cells. A significant increase in the transcript levels of LAMP1 gene in U18666A treated cells suggest that this compound might be modulating other endosomal sorting pathways.
4. Figure 4: The explanation that sustained exposure of lipid is required for upregulation of the NPC1 protein is very vague as there is a significant increase observed in the protein levels of NPC1 in H37Rv infected cells. The simultaneous inhibition and enhanced expression of NPC1 protein in Mtb infected macrophages needs to be reconciled by the authors.
5. Figure 5B: Again, unlike other lipid extracts, the fact that the NPC1 inhibitors significantly increase the overall cholesterol levels suggest that both the mechanism could possibly be using different pathways.

Thus one can argue that the lipids mediated cholesterol relocation in the cells could possibly be independent of NPC1 protein. Not sure why there is very distinct difference in the quality and clarity of the filipin stained images observed between the inhibitor treated positive control verses the lipid treated cells.

6. Why in particular, RAW264.7 cell line has been used throughout the study? Have the authors performed NPC1 inhibition by mycobacterial lipids in other cell lines as well, such as THP-1 or mouse derived primary cells – peritoneal macrophages or bone-marrow derived macrophages? It could be more convincing to know that the proposition holds true for all cell lines types. It could also further justify the specific use of RAW264.7 cell line.

7. Authors have failed to demonstrate hereby that blocking of NPC1 helped in persistence. A very casual use of term “Persistence” should be avoided because no such experimental evidence/ model has been shown to support the said claim. Furthermore, the authors have cited their previous publication where *M. smegmatis* and *M bovis* BCG have been used for intracellular survival. It may be noted that both are non-pathogenic, though BCG has the potential of NPC1 inhibition as evident in figure 2A. It is recommended that intracellular survival inside NPC1^{-/-} should be done with pathogenic *Mtb-Rv* rather than non-pathogenic strains.

8. Further, the said manuscript also lacked an understanding of how NPC1^{-/-} mice would survive if they were to be infected with certain representatives of Lineages used in Figure 1. To this end, it recommended a survival and bacterial burden (CFU) study be done in NPC1^{-/-} mouse till the chronic phase of infection and see whether the bacteria burden increase or decrease which could shed light on “persistence”.

9. It would be interesting to know exactly which component of crude lipid preparation is exhibiting NPC1 inhibitory activity. MS of the lipid could help identifying the active agent.

Minor comments:

1. Typo in LysoTracker spelling under result section.
2. At a certain location, *M. bovis* bacillus is written which should be corrected to *M bovis* Bacillus ... instead.

Reviewer #2 (Remarks to the Author):

NCOMMS-21-49975

The manuscript untitled, “Lipid mediated inhibition of Niemann-Pick C1 protein is an evolutionary conserved feature of multiple *Mycobacterium tuberculosis* lineages and non-tubercular mycobacteria” by Weng, Shepherd et al. investigate lipid-mediated modifications of endolysosomal fusion, emphasizing this study onto the role of NPC1 as primary target responsible for this phenotype.

The authors have combined the use of fluorescence-based assays, including FACS and immunofluorescence, NP-HPLC and MALDI lipid fingerprinting with immunoblot analysis to describe multiple cellular effects caused by mycobacterial lipid extracts. They show that lipid extracts from multiple mycobacterial species, including species from the Mtb complex or NTM species, are able to increase endolysosomal swelling and cholesterol hyper accumulation in RAW 264.7 mouse macrophages.

Understanding the molecular and cellular bases allowing mycobacteria to subvert and persist within their host is crucial in order to develop new therapeutic approach, therefore this study is of great interest for the entire scientific community in the TB field.

Despite containing some minor typos and sometimes a lack of precision, this manuscript is overall well-written. However, many of the findings that have been obtained and conclusions that have been drawn by the authors are not supported enough by the presented data. Indeed, some of the data are not strong enough, and require complementary approaches to robustly propose that NPC1 inhibition is caused by mycobacterial lipid extract, triggers global endolysosomal modifications and that this process is highly conserved among mycobacterial pathogens. In addition, providing new biological/molecular insights responsible for this phenotype such as identifying key mycobacterial/host lipids involved in this process. In their previous study, the authors combined multiple assays with live bacteria, heat-killed bacteria and purified lipid extracts to demonstrate that specific lipids induce modifications that are responsible for physiologically relevant phenotypes (e.g pathogenicity). The use of live strains and in cellulo infection models that complement the lipid extract data would be a real asset and defining whether the endolysosomal swelling promotes/restricts mycobacterial growth would really strengthen the manuscript.

Based on this lack of strong biological significance in the current version of the manuscript, I would recommend the editor to consider a transfer from Nature Communications to its sister journal Communications Biology.

Major comments:

Previous experiments performed with MAMES/FAMES or TDM by the same group to trigger NPC deficient phenotype were performed with 10 or 50 $\mu\text{g}/\text{mL}$ of lipid extracts for 18h (PMID: 28008422). Specially to see the sphingomyelin and GM1 Gangliosides phenotypes. The experiments carried out with mycobacterial lipid extracts in this study were presumably performed with 100 $\mu\text{g}/\text{mL}$ for 48h (mentioned Results section – Page 5). But in the legend of the Fig. 2 the author mentioned 50 $\mu\text{g}/\text{mL}$. Can the authors correct this discrepancy? Also update the material and method section accordingly and comment about this potential small change in between the two studies?

The authors use their own well-described assay (PMID: 25665453) to identify whether lipid extracts caused endolysosomal modifications, with special emphasize onto volume. For that they mainly rely onto LysoTracker staining and flow cytometry analysis. However, Mean LysoTracker Intensity per cell (or

singlets) cannot directly be used as readout for volume. Indeed, one cell can harbour few acidic compartments that display high-fluorescence levels whereas another one can harbour numerous acidic compartments with low-fluorescence levels. and these two cells might have a Mean LysoTracker Intensity that is similar. Can the author comment about this? I would strongly recommend to use an alternative approach with higher resolution such as fluorescence microscopy (as done in Fig3 or Fig5) allowing to reach subcellular resolution. Then acidic compartments might be easily identified as “puncta” and analysed by conventional quantitative imaging (intensity, size, number etc..).

Alongside LysoTracker-based experiments another complementary approach would be to use endolysosomal proteolytic/hydrolytic probes to determine whether increase of acidic compartment (number, size or volume) is associated with modification of proteolytic/hydrolytic function (PMID: 23253353) which might have important consequences for growth restriction/permissiveness.

The authors claimed that mycobacterial lipid extract treatment triggers expansion of the endolysosomal compartment in RAW macrophages using anti-LAMP-1 immunofluorescence. Quantitative analysis is essential to make such a claim. The authors cannot rely on one single representative image and must provide data showing that LAMP-1 positive compartments are substantially modified in 2D or in 3D. If the authors are not comfortable enough with subcellular fluorescence imaging, I will suggest to use EM as previously described for analysis of Mtb-mediated lysosomal swelling (PMID: 31427817). Investigating the LAMP-1 expression level (assessed by Q-PCR) or production level (assessed by immunoblot) was an important control but any modifications related to LAMP-1 production cannot be linked to lysosomal volume or function.

In Fig.5A, U18666A treatment triggers the formation of filipin positive puncta (from 5 to almost 100%). However, this treatment results in a less than 2-fold change (1.7-fold greater than vehicle) in cholesterol levels when assessed by Amplex red measurement of cholesterol. Can the authors comment about this? It is also worth mentioning that there is no script describing how the fluorescence microscopy analysis was performed (definition of a puncta, segmentation, counting etc..) for Fig 5A and Fig5B.

Filipin staining is pretty robust however might lack of specificity, I would recommend to stain as well with other conventional neutral lipid staining such as Bodipy or LipidTox staining and eventually look at TAG levels since not only cholesterol might be affected by this treatment.

The authors mainly claim by selective phenotypic analogy with the NPC1 inhibitor U18666A that mycobacterial lipid extracts will target NPC1 and therefore perturb cholesterol levels. But it is also not clear in this manuscript -especially in Fig4 and Fig5- why the authors are not describing in details when they have distinct results with the NPC1 inhibitor U18666A and with the lipid extract. Western-blot analysis showed that H37Rv lipid extracts slightly increase NPC1 protein levels, whereas U18666A and the other extracts did not. The filipin staining representative images and their characteristics are clearly different from U18666A treatment with the other conditions. That why describing how the identification

and analysis was carried out is essential. Can the authors comment about the number and size of puncta? The NPC1 inhibitor U18666A triggers a significant accumulation of cholesterol measured by amplex red quantification whereas it was not the case for any of the extract tested.

The authors need to carefully address these points and clarify some of the claim they made regarding lipid extract being NPC1 inhibitor by analogy with U18666A-mediated phenotypes.

Most of the concerns raised for the filipin staining approach in Fig.5 are also valid for the CtxB-GM1 experimental approach, specially analysis and interpretation. An accurate and detailed script describing how puncta were identified, counted and analysed must be provided as it is extremely difficult to define on these representative images, what is a puncta and what is not. The authors are also not mentioning the drastic changes in cell-shape and cytoplasmic volumes resulting from distinct treatments (UT/DMSO vs treated samples).

Glycosphingolipids (GSLs) extraction from RAW 264.7 and level determination by normal phase HPLC showed that different lipid extracts altered the level of GSLs. The authors claim that the GSLs accumulation is happening in all the conditions except with *M.canettii* cell wall extracts. However, GM2 & GA1 were significantly enriched in cells treated with *M.canettii* cell wall extracts. Can the authors comment about this?

If no significant changes in Mycobacterial lipid profiles were identified using MALDI fingerprinting, what would be the lipid present in *Mtb H37Rv* but absent in *M.canetti* resulting in this phenotype? Also, some NTM do not have specific lipids that are produced by the MTBC complex. Could the authors extensively discuss about these biochemical differences between *Mtb H37Rv* and NTM cell walls which are apparently not having huge impact onto the observed phenotype and conclusions. Have the authors considered doing specific lipid extraction to identified a potential candidate responsible for this phenotype, as done previously (PMID: 32142653)? Also How to be sure that the observed phenotype could not be due to one or several lipids present in all strains but expressed in various ratio ?

On a minor note, have the authors considered trying alternative cellular model to see whether this response and phenotype is conserved among different cell type? Such as mouse primary cells (BMDMs) or human cells such as THP-1 or hMDMs?

Reviewer #3 (Remarks to the Author):

The manuscript by Weng et al, entitled “Lipid mediated inhibition of Niemann-Pick C1 (NPC1) protein is an evolutionary conserved feature of multiple Mycobacterium tuberculosis lineages and non-tubercular mycobacteria” intends to describe the role of pathogenic mycobacteria lipids in inhibiting NPC1 pathways, which is shown to play a role in controlling the acidification of lysosomes, blocking phagosome maturation and thereby leads to mycobacteria persistence within host cells. Authors focused on *M. tuberculosis* strains from lineages 1, 2, and 4, and further claim that pathogenic mycobacteria evolved their cell wall lipids to block NPC1, where the *M. tuberculosis* complex ancestor strain *M. canettii* is not able to block NPC1, leading to the assumption that “the evolution of mycobacterial cell wall lipids that inhibit the NPC pathway evolved early and post divergence from *M. canettii*-related mycobacteria and that NPC1 inhibition significantly contributes to the ability of these pathogens to persist and cause disease”. This is an interesting and important study, the major issue is that it seems too premature for publication and that many of the strong statements in the abstract, results and discussion section are not supported well by the data and/or are too speculative.

Major Concerns:

1/ As the authors recognize in their discussion, although a phenotype is observed related to NPC1, it is not clear what drives this phenotype, as major differences in lipids were not observed. Further, as author discussed, it is not clear if the phenotype observed is due to an enrichment or differential recovery of ‘active’ lipids species during the extraction process. This doubt is enhanced as different strains within the lineages studied have different phenotypes, especially for lineages 1 and 2.

2/ Although the observations are interesting, there are some concerns about how data are presented/interpreted and mainly, it is not clear the mechanism(s) behind the phenotype observed, especially as not significant differences between the lipids studied can justify the observed phenotype. In this regard, it is unclear why *M. canettii* lipids were not studied by TLC, especially as this is the strain which lipids do not induce the NPC pathway.

3/ In addition, the claim that “lipid mediated inhibition of NPC1 protein is an evolutionary conserved feature of multiple Mycobacterium tuberculosis lineages and non-tubercular mycobacteria” needs to be carefully evaluated, as few strains are studied to claim this strong (and definitive) statement, especially when even lineage 1 and 2 present *M. tuberculosis* strains which lipids do not show the entitled phenotype. Further, the addition of recently described *M. tuberculosis* phylogenetically closer to *M. canettii* (state in South India) with 99% identity could be of interest to support this premise, as well as adding closer relatives to *M. canettii* in the phylogenetic tree, such as *M. lacus*, *M. shinjukuense*, and *M. riyadhense*.

4/ The phenotype observed needs to be evaluated by dose-titrations of the lipids as well as in different cells to be certain that is not a cell specific phenomena.

5/ Authors also found that there was not an increase in the production of LAMP-1, but instead a redistribution of the protein, but this fact is not demonstrated.

6/ Previously, authors reported that infection of RAW 264.7 MF with *M. bovis* BCG causes a significant upregulation of NPC1 protein; however, this is not clearly observed when using lipids from different *M. tuberculosis* strains, where a sustained exposure to lipids may be required. How realistic is this? Little is known about the biosynthesis and restructuration of the *M. tuberculosis* cell wall during infection, which could be also host cell dependent.

7/ It seems that NPC1 is not the full story, may be part of the story, and thus better correlations need to be presented. For example, is there a correlation between *M. tuberculosis* intracellular growth of the different strains studied and the levels of inhibition for NPC1?

8/ Authors need to consider the existence of other cell wall components, such as lipoglycans and glycosylated-lipoproteins that are not being studied here and also directly contribute to *M. tuberculosis* survival within host cells by inducing phagosome maturation arrest, this is not being studied, either mentioned. In addition, some of the major regulators of the faith of *M.tb* within macrophages are the mannose-containing lipoglycans. How the presence of these compounds mask/enhance the effects of the lipids as shown here?

9/ It is not clear why other lineages, especially lineage 3 was not studied, especially in the light that strain in lineage 3 are causing drug resistance TB outbreaks.

10/ There are some overstatements generalizing the results, when authors only studied few *M.tb* strains in the NPC pathway experiments. The use of NTM and *M. smegmatis*, as well as other *M. tuberculosis* strains lipids, was only performed in the acidification study.

11/ There is not a direct read out of NPC1 activity being diminished. Studies using the inhibitor plus the lipids will add a light into demonstrating if lipids further enhance NPC1 inhibition or using NPC1 activity stimulants plus the mycobacterial lipids studied.

12/ The extraction of lipids used will extract lipooligosaccharides (LOS, polar lipids); however, the resuspension of the extracted lipids with only chloroform will make difficult the analysis of LOS. In addition, the argument that *M. smegmatis* does not induce NPC1 because of the presence of differential lipids such as non-serovar specific glycopeptidolipids (GPLs) fails in concept, because strains of the *M. avium* complex seem to induce NPC1 and have both non- serovar and serovar specific GPLs.

13/ Fig. 2: Acidification induced by total lipids happens for lineage 4 (100% of the strains tested, 7/7), lineage 1 (60% of the strains tested (3/5)) and lineage 2 (67% of the strains tested (2/6) but with very heterogeneous MFIs values. Of these, authors only choose one strain from each lineage to determine the effects of the lipids on the NCP pathway vs. *M. canettii* and *M. tuberculosis* H37Rv. Although reasonable, this selection limits the impact of their claims; and thus, one cannot extrapolate their results to the general statements that authors mention in the results, discussion and abstract (as well as title).

14/ Fig 3: Need quantifications. By flow, UT, DMSO, *M. canettii* show similar results; however, by confocal looking at LAMP-1 these results differ, where *M. canettii* or DMSO seem higher than even the positive control U18666A. In this regard, the positive control appears similar to untreated cells. Further, when comparing conditions/strains with similar values [as obtained by flow (MFI)], in some cases these differ by confocal microscopy (e.g. UT vs. 639). Further, another marker of lysosome accumulation such as CD63 will be necessary to confirm results. As well as other type of cells. It is not clear how many times the experiment in Fig. 3 Panel A was done. Also, quantifications are required as it seems that DMSO produces higher accumulation of LAMP-1 than the positive control.

15/ Fig 4: Irrelevant if the impact is on NPC1 activity (acidification and accumulation of lysosomes) and not on NPC1 levels. Further, which *M. tuberculosis* H37Rv lipids are the ones tested here (BEI or in house obtained)? *M. canettii* in the blot presented in Fig. 4 looks less than DMSO but the quantification shows similar results. A better representative blot may be needed. Actually, the beta-actin in the blot for *M. canettii* seems higher so the value of the NPC1 should be event lower when normalized to beta-actin. It is not clear why the U18666A control is added, are the authors expecting a decrease in levels, activity or both? And if it is a decrease in activity, this is not directly well proven.

16/ Fig. 5: Fig. 5A and B can be combined. It is not clear from the selected pics that *M. canettii* has less puncta than *M. tuberculosis* H37Rv. Some regions marked by arrows in *M. tuberculosis* H37Rv look similar to *M. canettii* but these are not marked. Authors need to clarify what they chose to mark or add representative images of their findings. Further, authors state that these are lysosomes containing cholesterol. Proper markers will need to be used to show co-localization of lysosomes with cholesterol. Graphs in 5Ci and 5Cii can be combined, the issue here is that any of the lipids extracts induce cholesterol accumulation at the levels observed by U18666A. This does not match the results obtained in 5B.

It is not clear if this is lysosomal or endosomal accumulation, this needs to be discerned to make this claim. Co-localization of Filipin with LAMP-1 or CD63 will be required. It will be of interest to show if the accumulation of LDL-vesicles also occurs as shown for NPC, and what this will mean.

17/ Fig 6: Representative figures need to be chosen carefully. As in previous figures, *M. canettii* seems to have more puncta than *M. tuberculosis* 333. Images provided seem that do not represent the quantification presented. In addition, authors need to revise the scale, because although in the figure the scale is the same, cells in *M. canettii* seem (perception) smaller than in the other groups. Also authors need to verify that accumulation happened in lysosomes vs. early/late endosomes.

18/ Fig 7: There are some quantifications that show significance in GM populations after exposure to the lipids of some *M. tuberculosis* strains vs. *M. canettii* but the HPLC plots as presented do not show these significant differences. For comparative reason, graphs in Fig. 7C need to be in the same scale.

19/ Fig S3: This figure needs to be labeled and explained better. In panel E some strains in this panel are irrelevant to the presented study, and add confusion showing not clear lipid differences between strains that cause or do not cause the NPC1 phenotype; thus, without defining an obvious link.

20/ Fig S3A: There are differences in the levels of some of the lipids on the strains evaluated in this study. For example, strain 333 has less DIM B but more TAG than the other 2 strains (281 and 639). Same for some of the PIMs and TMM.

21/ Fig S3B: Again, differences in lipids are shown in supplemental Figure S3. Correlations between the presence/abundance of a specific lipids and the phenotypes observed will be important. This study is also no accounting the expression of these lipids during infection as well as other cell wall components present in the cell envelope that are not in the lipid fraction.

In addition, a difference between *M. canettii* and other MTBC strains is the presence of phenolic glycolipid, present in HN878 and *M. bovis*. This could be experimentally addressed.

22/ Dispersed soluble lipids vs. in micellar form, did the authors try different amounts of lipids to see if the phenomena of micelle formation plays a role? is there a total lipid dose response?

Minor:

23/ Fig. 1 could be moved to supplement, and the phylogenetic tree used based on the expression of HupB needs to be better justified in the introduction.

24/ In some experiments/figures it is not clear how many cells/events were counted.

25/ "TB is a pathogen". This is inaccurate as stated and needs to be revised.

26/ Table 2: Mean +/- SEM needs to be presented in this table.

Authors responses to reviewers' comments.

First, we wish to thank the reviewers for their extremely helpful reviews and comments on our manuscript

We will address in turn the points raised by the three reviewers and will indicate the changes that we have made to the manuscript and where new data have been included.

Responses to points raised by reviewer 1.

The word “persist” can only be used in the context of “long-term disease persistence” or “antibiotic persistence” (Balaban et al., 2019, Nature Reviews Microbiology 17, 441-448) neither of which has been demonstrated to be modulated by NPC1 inhibition. Hence use of the word “survive” sounds more appropriate.

We agree that the term “persist” or “persistence” should only be used in specific contexts and have therefore replaced with the word “survive” in the manuscript where appropriate.

Major comments:

2. Figure 2: The fact that not all the strains (2/5) for L1 and (2/6) for L2 adhere to the said hypothesis, attributed to the potency and variation in lipid components between strains, can only be reconciled by identifying the active component(s) directly modulating NPC1 function.

As the reviewer indicated, when we tested the lipid extracts from five strains representative of lineage 1 at a lipid concentration of 100 µg/ml, 2 of them did not increase LysoTracker™ staining significantly (2/5) and similarly, 2 of the strains from lineage 2 also did not reach significance (2/6). We hypothesized that the explanation for this was that the concentration of the specific biologically active lipid in those samples was below the threshold when tested at 100 µg/ml. We have therefore now tested those specific *Mtb* strains (plus positive and negative controls) at 200 µg/ml and we were able to confirm a significant increase in LysoTracker™ staining. Therefore, the lipid extracts from all of the *Mtb* strains representative of lineages 1, 2 and 4 that we sampled significantly increase LysoTracker™ staining, thereby demonstrating the widespread distribution of this activity. We have included these additional data as Fig 1C and modified the text in the results section accordingly.

We have also expanded our analysis to include a lipid extract from the *Mtb* strain (91_0079) from lineage 3 and confirmed significant activity, thereby extending evidence of the presence of activity to now include this lineage for the first time (Fig 1C).

3. Figure 3: Quantification is missing. The authors should show a larger field of interest. The conclusion that “LAMP1 immunoreactivity was higher in lipid-treated cells...” is debatable and I do not see any difference in the immunoreactivity to LAMP1 antibodies in any of the panels except for the U186660 treated cells. A significant increase in the transcript levels of LAMP1 gene in U18666A treated cells suggest that this compound might be modulating other endosomal sorting pathways.

As requested by the reviewer we have prepared new microscopy images of our data and have included larger fields of interest, along with those at higher magnifications (Fig 2 and 3) to illustrate the increase in lysosome size.

We have also made quantified the LysoTracker™ positive and immunoreactive (LAMP-1) compartments and confirmed a significant increase in both mean late endosome/lysosomal size and size distribution in U18666A and *Mtb* lipid treated cells but not in cells incubated with the *M. canettii* extract (Fig 2C and D, Fig 3C and D).

4. *Figure 4: The explanation that sustained exposure of lipid is required for upregulation of the NPC1 protein is very vague as there is a significant increase observed in the protein levels of NPC1 in H37Rv infected cells. The simultaneous inhibition and enhanced expression of NPC1 protein in Mtb infected macrophages needs to be reconciled by the authors.*

We should clarify that there is extensive evidence in the literature that drugs which block NPC1 activity leads to increased NPC1 protein levels^{1,2} via a mechanism that has not been precisely elucidated.

5. *Figure 5B: Again, unlike other lipid extracts, the fact that the NPC1 inhibitors significantly increase the overall cholesterol levels suggest that both the mechanism could possibly be using different pathways. Thus one can argue that the lipids mediated cholesterol relocation in the cells could possibly be independent of NPC1 protein. Not sure why there is very distinct difference in the quality and clarity of the filipin stained images observed between the inhibitor treated positive control verses the lipid treated cells.*

U18666A is a potent inhibitor that gets efficiently into cells, builds up in the lysosome and rapidly inhibits NPC1 function. The mycobacterial lipids are a complex mixture containing one or more NPC1 inhibitory lipids and are not surprisingly therefore less potent than the pure U18666A inhibitor.

In order to improve the quality of the images and clarity of cholesterol accumulation as revealed by filipin staining we have converted the microscopy images to black and white to aid visual inspection. This format has been successfully employed in multiple published studies of filipin staining of cholesterol accumulation in the endo-lysosomal system for the same reason³.

The changes to cholesterol distribution and total cellular level of the lipid that we show are fully consistent with the published literature for NPC. The endo-lysosomal accumulation of cholesterol is one of the defining characteristics of NPC and indeed was for many decades the basis for clinical diagnosis in skin fibroblasts from patients¹. Similarly, increased amounts of cholesterol have been reported in patient tissues and cells^{4,5}, in all animal models of NPC⁶ and in U18666A treated cells⁷.

6. *Why in particular, RAW264.7 cell line has been used throughout the study? Have the authors performed NPC1 inhibition by mycobacterial lipids in other cell lines as well, such as THP-1 or mouse derived primary cells – peritoneal macrophages or bone-marrow derived macrophages? It could be more convincing to know that the proposition holds true for all cell lines types. It could also further justify the specific use of RAW264.7 cell line.*

We apologise for not fully explaining the selection of cell line (RAW 264.7) as this was based on our previous paper⁸ which provided evidence that pathogenic mycobacteria and lipid extracts induced the identical phenotypes in human monocyte-derived MΦ). We have modified the text in the introduction as follows:

“The precise function of NPC1 is currently unclear and a direct functional assay of its activity is not available. We therefore demonstrated that cells harbouring specific mycobacteria display the unique combination of phenotypes that result from loss of NPC1 activity¹². In brief, all of the cellular phenotypes that in combination define NPC, including a reduction in the calcium content of lysosomes, prevention of phagosome-lysosome fusion, accumulation of specific lipid species (including cholesterol, sphingomyelin and glycosphingolipids (GSLs)) within the endo-lysosomal system were induced in cells infected with mycobacteria that survive within host cells (*Mtb* and *Mycobacterium bovis bacillus Calmette-Guerin* (BCG)), but not the environmental non-pathogenic mycobacterium *Mycobacterium smegmatis*. Very significantly, it was the lipid fraction of the cell wall from the pathogenic mycobacteria that inhibited the NPC pathway. Critically, all of the NPC cellular phenotypes were induced not only in infected and lipid-treated murine RAW 264.7 MΦ but also in human, monocyte-derived MΦ, thereby confirming relevance to human disease”.

7. Authors have failed to demonstrate hereby that blocking of NPC1 helped in persistence. A very casual use of term “Persistence” should be avoided because no such experimental evidence/ model has been shown to support the said claim. Furthermore, the authors have cited their previous publication where *M. smegmatis* and *M bovis* BCG have been used for intracellular survival. It may be noted that both are non-pathogenic, though BCG has the potential of NPC1 inhibition as evident in figure 2A. It is recommended that intracellular survival inside NPC1^{-/-} should be done with pathogenic *Mtb-Rv* rather than non-pathogenic strains.

We thank the reviewer for raising these points. As requested, we have replaced the term “persistence” with “survival” throughout the manuscript. As indicated above, we have added text to restate the findings in the study by Fineran *et al*⁸ that showed that *Mtb* had the same activity and a lipid extract of the mycobacterium induced all of the NPC phenotypes seen with BCG.

We agree that it would be of interest to explore the infection and survival of mycobacteria within NPC1^{-/-} cells and mice. However, such studies as suggested are beyond the scope of this publication and are likely to be technically challenging, especially *in vivo*. For example, *Npc1*^{-/-} mice have a significantly shortened lifespan (10-12 weeks) and undergo progressive disease and neurodegenerative decline.

8. Further, the said manuscript also lacked an understanding of how NPC1^{-/-} mice would survive if they were to be infected with certain representatives of Lineages used in Figure 1. To this end, it recommended a survival and bacterial burden (CFU) study be done in NPC1^{-/-} mouse till the chronic phase of infection and see whether the bacteria burden increase or decrease which could shed light on “persistence”.

We refer the reviewer to our response to point 7 above.

9. It would be interesting to know exactly which component of crude lipid preparation is exhibiting NPC1 inhibitory activity. MS of the lipid could help identifying the active agent.

We agree with the reviewer that identification of the active lipid is of interest and is something we are actively pursuing. However, as the reviewer will also likely understand, this is not a trivial task and involves the large-scale growth, systematic extraction, fractionation, testing, analysis and re-verification of a large quantity of mycobacteria, which cannot be prepared as a single culture of sufficient size because

of legal restrictions on volume. It is beyond the scope of this manuscript and is currently in progress.

Minor comments:

1. Typo in LysoTracker spelling under result section.

Corrected

2. At a certain location, *M. bovis bacillus* is written which should be corrected to *M bovis Bacillus* ... instead.

Corrected.

Reviewer #2 (Remarks to the Author):

Major comments:

Previous experiments performed with MAMES/FAMES or TDM by the same group to trigger NPC deficient phenotype were performed with 10 or 50 µg/mL of lipid extracts for 18h (PMID: 28008422). Specially to see the sphingomyelin and GM1 Gangliosides phenotypes. The experiments carried out with mycobacterial lipid extracts in this study were presumably performed with 100 µg/mL for 48h (mentioned Results section – Page 5). But in the legend of the Fig. 2 the author mentioned 50 µg/mL. Can the authors correct this discrepancy? Also update the material and method section accordingly and comment about this potential small change in between the two studies?

The sources of lipids used in our previous study⁸ were different (commercial or gift from academic lab generating lipid fractions relevant to CD1d lipid presentation) from those used in this study. In order to exclude variability as far as is possible we decided to test lipids and other extracts from two verified sources; BEI, for which analysis data are available for the specific batch of each reagent; lipid extracts prepared in the laboratory of the co-author Dr G. Larrouy-Maumas with whom we are actively collaborating. Indeed, to confirm comparable activity in extracts from the two sources we compared the H37Rv extracts from the two sources and found similar levels of activity (Supplementary Fig 2A and C).

We apologise for the discrepancy between the lipid concentration stated in the main text and the figure legend, which was made in error. We have corrected the legend to Fig 1A to indicate the concentration was 100 µg/ml. Note – in response to a point also made by reviewer 1 concerning the lack of activity in two *Mtb* strains from lineage 1 and two from lineage 2, we performed and now include results for analysis of cells treated with 200 µg/ml lipid from these *Mtb* strains, as well as controls, including *M. canettii*. We show that at this higher concentration all four *Mtb* strains display significant activity, whereas *M. canettii* is inactive (Fig 1C).

The authors use their own well-described assay (PMID: 25665453) to identify whether lipid extracts caused endolysosomal modifications, with special emphasize onto volume. For that they mainly rely onto LysoTracker staining and flow cytometry analysis. However, Mean LysoTracker Intensity per cell (or singlets) cannot directly be use as readout for volume. Indeed, one cell can harbour few acidic compartments that display high-fluorescence levels whereas another one can harbour numerous acidic compartments with low-fluorescence levels. and these two cells might have a Mean LysoTracker Intensity that is similar. Can the author comment about this?

As described previously in our paper that delineated the methodology⁹ we were careful to define the intensity of LysoTracker™ staining as determined by FACS as “a measure of the relative late endosome/ lysosome volume of cells” (page 5, line 8) and agree that equivalent values could be obtained from cells that contain a smaller number of larger lysosomes and cells with more lysosomes of individually smaller volume. Therefore, using this technique LysoTracker™ staining intensity is indicative of the relative size of the total acidic compartment of the cell.

In order to extend the data, we now include quantification of LysoTracker™ staining visualised by microscopy (Fig 2) and confirm a statistically significant increase in the number and volume of individual lysosomes in cells exposed to *Mtb* lipids (Fig 2). Furthermore, we have also quantified the volume of late endosomes/lysosomes via inspection of LAMP-1 staining (Fig 3) and found a comparable increase in size and size distribution with the specific treatments.

Alongside LysoTracker-based experiments another complementary approach would be to use endolysosomal proteolytic/hydrolytic probes to determine whether increase of acidic compartment (number, size or volume) is associated with modification of proteolytic/hydrolytic function (PMID: 23253353) which might have important consequences for growth restriction/permissiveness.

We agree that it would be of interest to determine whether *Mtb* lipid-mediated inhibition of NPC1 may also affect the degradative activities of lysosomes and thereby also permissiveness. Our intention in this manuscript was to provide evidence of the widespread distribution of NPC1 inhibitory lipid(s) amongst different pathogenic mycobacteria. Whether inhibition of NPC1 also results in other cellular changes that favour the establishment of intracellular mycobacteria is an area that we plan to address in future investigations.

*The authors claimed that mycobacterial lipid extract treatment triggers expansion of the endolysosomal compartment in RAW macrophages using anti-LAMP-1 immunofluorescence. Quantitative analysis is essential to make such a claim. The authors cannot rely onto one single representative image and must provide data showing that LAMP-1 positive compartments are substantially modified in 2D or in 3D. If the authors are not comfortable enough with subcellular fluorescence imaging, I will suggest to use EM as previously described for analysis of *Mtb*-mediated lysosomal swelling (PMID: 31427817). Investigating the LAMP-1 expression level (assessed by Q-PCR) or production level (assessed by immunoblot) was an important control but any modifications related to LAMP-1 production cannot be linked to lysosomal volume or function.*

We agree that quantitative analysis is necessary to substantiate that *Mtb* lipids expand the endolysosomal compartment, as defined by LAMP-1 immunostaining. We now include microscopy data from both LysoTracker™ and anti-LAMP-1 staining (Fig 2 and 3) and quantitative analysis which confirms a statistically significant increase in the compartment (diameter of LysoTracker™ positive and LAMP-1 positive compartments) in cells exposed to multiple *Mtb* lipids, but not to *M. canettii* (Fig 2 and 3). However, we were unable to measure a significant increase in LAMP-1 transcripts or protein with exposure to lipids, suggesting to us that the immunoreactivity in the expanded compartments reflects re-distribution of the same amount of LAMP-1 protein.

In Fig. 5A, U18666A treatment triggers the formation of filipin positive puncta (from 5 to almost 100%). However, this treatment results in a less than 2-fold change (1.7-fold greater than vehicle) in cholesterol levels when assessed by Amplex red measurement of cholesterol. Can the authors comment about this?

Again, we thank the reviewer for questioning this apparent discrepancy between intensity of filipin staining and the extent of cholesterol change as assessed by Amplex Red quantification (Fig 5). This observation is widespread in the NPC literature^{1,7}. A plausible explanation is that recognition of cholesterol by filipin is influenced significantly by the local lipid membrane microenvironment in which it resides, thereby intensifying the signal. On the other hand, the Amplex Red assay quantifies cholesterol biochemically. As has been reported, there are limitations with all of the fluorescent reporters currently used for determining cellular distribution of cholesterol¹⁰.

It is also worth mentioning that there is no script describing how the fluorescence microscopy analysis was performed (definition of a puncta, segmentation, counting etc..) for Fig 5A and Fig5B.

This has now been included in Materials and Methods.

Filipin staining is pretty robust however might lack of specificity, I would recommend to stain as well with other conventional neutral lipid staining such as Bodipy or Lipidtox staining and eventually look at TAG levels since not only cholesterol might be affected by this treatment.

Filipin staining to detect endo-lysosomal accumulation of cholesterol in NPC has been used almost universally and reported in the NPC literature, is a defining cellular feature of the disease and has been the basis for clinical diagnosis¹. Here, use of this technique was made to confirm induction of one of the hallmarks of NPC in mycobacterial lipid treated cells. Lipidtox has been tested in NPC1 patient cells⁹ and was not different from controls.

The authors mainly claim by selective phenotypic analogy with the NPC1 inhibitor U18666A that mycobacterial lipid extracts will target NPC1 and therefore perturb cholesterol levels. But it is also not clear in this manuscript -especially in Fig4 and Fig5- why the authors are not describing in details when they have distinct results with the NPC1 inhibitor U18666A and with the lipid extract. Western-blot analysis showed that H37Rv lipid extracts slightly increase NPC1 protein levels, whereas U18666A and the other extracts did not. The filipin staining representative images and their characteristics are clearly different from U18666A treatment with the other conditions. That why describing how the identification and analysis was carried out is essential. Can the authors comment about the number and size of puncta? The NPC1 inhibitor U18666A triggers a significant accumulation of cholesterol measured by amplex red quantification whereas it was not the case for any of the extract tested.

The authors need to carefully address these points and clarify some of the claim they made regarding lipid extract being NPC1 inhibitor by analogy with U18666A-mediated phenotypes.

U18666A was included as a positive control for induction of NPC cellular phenotypes. This compound has been used extensively, it is able to induce all of the phenotypes and has been demonstrated to bind directly to NPC1¹¹. As explained above, we have clarified in the text that in the absence of a direct assay for NPC1 activity, NPC is defined by the expression of a unique combination of cellular phenotypes that result from loss of NPC1 function rather than the magnitude of individual ones. Lipid extracts

representative of the three *Mtb* lineages, but not *M. canettii* induced all of them, but to differing extents. At this point we can only speculate as to reasons why the magnitude of effects observed are not absolutely identical, but would suggest that varying amounts of bioactive lipid in the *Mtb* extracts, differing properties between lipids and U18666A (the latter is lysosotropic) may be responsible. For example, whilst filipin staining reveals a dramatic re-distribution of cholesterol within NPC cells, the absolute amounts of the lipid may not be significantly changed (see above).

We have included a definition for the identification of filipin puncta to aid inspection of images in Materials and Methods and their quantification.

In respect of apparent differences in magnitude of changes in cholesterol as visualized by filipin staining and determined by Amplex Red, please see the response above.

Most of the concerns raised for the filipin staining approach in Fig.5 are also valid for the CtxB-GM1 experimental approach, specially analysis and interpretation. An accurate and detailed script describing how puncta were identified, counted and analysed must be provided as it is extremely difficult to define on these representative images, what is a puncta and what is not.

We have included a definition for the identification of Ctxb staining that localises to the endo-lysosome to aid inspection of images and detailed the method of analysis and quantification.

The authors are also not mentioning the drastic changes in cell-shape and cytoplasmic volumes resulting from distinct treatments (UT/DMSO vs treated samples).

We frequently observed apparent changes to cell shape following lipid treatments. Interestingly, we could not confirm a significant change in size (determined by forward scatter) when cells were analysed by FACS, but this could reflect the requirement to detach cells from surfaces for analysis. Because observation suggested that such changes in morphology varied in respect of their extent, we consider that it will be better addressed when experiments can be undertaken using defined amounts of pure lipids

*Glycosphingolipids (GSLs) extraction from RAW 264.7 and level determination by normal phase HPLC showed that different lipid extracts altered the level of GSLs. The authors claim that the GSLs accumulation is happening in all the conditions except with *M.canettii* cell wall extracts. However, GM2 & GA1 were significantly enriched in cells treated with *M.canettii* cell wall extracts. Can the authors comment about this?*

We have made changes to the text describing these results (see below) to give more detail of the quantitative changes in specific lipid abundance in response to exposure to each mycobacterial lipid extract to highlight that all *Mtb* strains increased the levels of the most prominent lipid species (GM1a, GM1b and GD1a) whereas the response to *M. canettii* was significantly different, as illustrated by the reduction in GM1b and GD1a. Whilst we acknowledge that *M. canettii* increased the abundance of GM2 and GA1, the increase was much lower than seen with the *Mtb* strains and these two lipid species are of low abundance. We have made the following change to the text:

“Lipid extracts from all four *Mtb* strains also significantly increased total GSL levels in treated MΦ, but in contrast, total GSLs were significantly lower in cells incubated with

M. canettii lipids (Fig 7D). Levels of the most abundant lipid species GM1a, GM1b and GD1a were significantly increased by lipid extracts from all *Mtb* strains but in the case of *M. canettii* were either not significantly changed (GM1a) or were significantly lower (GM1b and GD1a). Strain 281 from *Mtb* lineage 1 significantly increased eight of the individual lipid species, lineage 2 strain 33 increased five lipid species, and lineage 4 strains 639 and H37Rv increased four and seven distinct lipid species respectively (Fig 7D and Table 2). *M. canettii* cell wall lipid extract only significantly enhanced the amounts of two minor lipid species (GM2 and GA1), albeit to a relatively small extent, did not alter the level of three individual lipids (GA2, Gb3 and GM1a) but significantly reduced the level of four lipids (Lac, GM3, GM1b and GD1a (Table 2), which was not observed with any of the *Mtb* strains”.

If no significant changes in Mycobacterial lipid profiles were identified using MALDI fingerprinting, what would be the lipid present in Mtb H37Rv but absent in M.canetti resulting in this phenotype? Also, some NTM do not have specific lipids that are produced by the MTBC complex. Could the authors extensively discuss about these biochemical differences between Mtb H37Rv and NTM cell walls which are apparently not having huge impact onto the observed phenotype and conclusions.

We thank the reviewer for pointing out this very important observation and comment. We have now amended the text accordingly:

“More specifically, based on the evolution of MTBC due to genetic deletions or rearrangement of regions of the genome that contains enzymes involved in the synthesis of surface exposed lipids and glycolipids, some lipids found in *M. canettii* cannot be produce by *Mtb* H37Rv¹²⁻¹⁶. Indeed, through evolution, *Mtb* has lost the ability to produce lipooligosaccharide because of *pks5* locus recombination¹⁷. With respect to the biochemical differences between *Mtb* H37Rv and NTM, based upon the literature, we can identify diglycosylated glycopeptidolipids and triglycosylated glycopeptidolipids which are known to make up more than 70% of the surface-exposed mycobacterial lipids in *M. abscessus* and are not found in *Mtb* H37Rv¹⁸⁻²². In addition, mycolic acids, such as ones bound to trehalose in TMM and TDM, and more particularly their decorations can differ between mycobacterial species^{23,24}. Indeed, mycolic acids display high structural diversity with variations in chain length (60 to 90 carbon atoms), the extent of unsaturation, and the chemical groups, such as ketones and methoxys²⁴. This high degree of diversity provides mycolic acids with species-specific characteristics²⁵. In addition to mycolic acids, other lipids are also specific to particular species of mycobacteria. For example, sulfolipid 1 and polyacyltrehalose are found exclusively in *M. tuberculosis*, while trehalose polyphleate is present in NTM species²⁶⁻²⁸. Layre et al²⁹. screened lipids that are present in *M. tuberculosis* but not in *M. bovis* BCG by an HPLC–MS-based lipidomics platform and identified 1-tuberculosinyladenosine (1-TbAd), an amphipathic diterpene nucleoside but its presence or absence in NTM have not been reported as yet.”

In addition, it is worth mentioning that in our study, we used MALDI fingerprinting, which gives a broad overview of the lipid composition as mainly the most ionizable lipids will be detected and therefore, more advance methods such as LC/MS should be employed in order to determine the detailed discrepancy between the composition and abundance of extractible lipids between *Mtb* H37Rv and NTM.

Have the authors considered doing specific lipid extraction to identify a potential candidate responsible for this phenotype, as done previously (PMID: 32142653)? Also How to be sure that the observed phenotype could not be due to one or several lipids present in all strains but expressed in various ratios?

We agree with the reviewer that identification of the active lipid is of interest and is something we are actively pursuing. However, as the reviewer will also likely understand, this is not a trivial task and involves growth, systematic extraction, fractionation, testing, analysis and re-verification of a large quantity of mycobacteria, which cannot be prepared as a single culture of sufficient size because of legal restrictions on volume.

As suggested by the reviewer, it is a possibility that activity results from a combination of specific lipids that are present in differing ratios in particular *Mtb* strains. This would be a question we expect to resolve when the identity of the bioactive lipid(s) has been confirmed.

On a minor note, have the authors considered trying alternative cellular models to see whether this response and phenotype is conserved among different cell types? Such as mouse primary cells (BMDMs) or human cells such as THP-1 or hMDMs?

This point was also raised by reviewer 1. (Point 6). Indeed, such data were included in our previous publication⁸ confirming that identical cellular phenotypes are seen in human monocyte-derived macrophages infected with specific mycobacteria and *Mtb* lipid extracts.

Reviewer #3 (Remarks to the Author):

Major Concerns:

1/ As the authors recognize in their discussion, although a phenotype is observed related to NPC1, it is not clear what drives this phenotype, as major differences in lipids were not observed. Further, as the author discussed, it is not clear if the phenotype observed is due to an enrichment or differential recovery of 'active' lipid species during the extraction process. This doubt is enhanced as different strains within the lineages studied have different phenotypes, especially for lineages 1 and 2.

We thank the reviewer for highlighting this question. Our objective in this manuscript was to investigate the distribution of lipid-mediated inhibition of NPC1 amongst *Mtb* clinical strains, as well as other important tubercular and non-tubercular species, thereby underscoring the importance of the mechanism to survival of mycobacteria. Interestingly, our new data showing that we were able to measure significant activity in the extracts of the four strains from lineages 1 and 2 (232, 346, 212 and 374) when tested at the higher concentration of 200 µg/ml (Fig 2C) is consistent with differing amounts of specific lipid(s) between the extracts. However, at this stage we cannot determine absolutely whether this reflects different lipid levels in distinct *Mtb* strains or is the result of differential recovery. We would expect to be able to address this issue when the bioactive lipid(s) has been identified (in progress).

2/ Although the observations are interesting, there are some concerns about how data are presented/interpreted and mainly, it is not clear the mechanism(s) behind the phenotype observed, especially as not significant differences between the lipids studied can justify the

observed phenotype. In this regard, it is unclear why *M. canettii* lipids were not studied by TLC, especially as this is the strain which lipids do not induce the NPC pathway.

We thank the reviewer for the comment and suggestion. We have now amended the main text accordingly in the discussion regarding the differences between *Mtb* and *M. canettii*. Regarding the *M. canettii* TLC, the main difference is in the production of LOS which several studies have reported³⁰ and we wished to focus primarily on the difference between *Mtb* lineages.

3/ In addition, the claim that “lipid mediated inhibition of NPC1 protein is an evolutionary conserved feature of multiple *Mycobacterium tuberculosis* lineages and non-tubercular mycobacteria” needs to be carefully evaluated, as few strains are studied to claim this strong (and definitive) statement, especially when even lineage 1 and 2 present *M. tuberculosis* strains which lipids do not show the entitled phenotype. Further, the addition of recently described *M. tuberculosis* phylogenetically closer to *M. canettii* (state in South India) with 99% identity could be of interest to support this premise, as well as adding closer relatives to *M. canettii* in the phylogenetic tree, such as *M. lacus*, *M. shinjukuense*, and *M. riyadhense*.

As explained above, we have re-tested the four *Mtb* strains within lineages 1 and 2 that previously were inactive at a lipid concentration of 100 µg/ml and provide new data confirming significant activity when tested at 200 µg/ml (Fig 2C).

To expand the breadth of strains examined, we have now included an extract of a representative of lineage 3, which showed significant activity (Fig 2C). This was the only strain from lineage 3 available from BEI.

We thank the reviewer for bringing to our attention the existence of additional mycobacterial species that may be of interest. Unfortunately, we do not currently have immediate access to the species mentioned, which if available would also require culturing and preparation of extracts. However, we will certainly test them if we can obtain them in future investigations.

4/ The phenotype observed needs to be evaluated by dose-titrations of the lipids as well as in different cells to be certain that is not a cell specific phenomena.

We have now included data to demonstrate that the effect of *Mtb* lipids on LysoTracker™ intensity is dose dependent (Fig S3C and D).

As mentioned above, we confirmed in our previous publication⁸ that identical NPC phenotypes are induced in both RAW 264.7 cells and human blood monocyte-derived macrophages. We have clarified this point in the introduction.

5/ Authors also found that there was not an increase in the production of LAMP-1, but instead a redistribution of the protein, but this fact is not demonstrated.

As shown in Fig 3 we used LAMP-1 staining to identify late endosome/lysosome compartments and by quantitation have confirmed an expansion in size of the LAMP-1 positive compartments in *Mtb* lipid treated cells (Fig 3C and D). However, quantification of both LAMP-1 transcripts by Q-PCR and protein by western blotting did not confirm significant increases in either after *Mtb* lipid treatment (Fig 3 E and F). We therefore suggest that the LAMP-1 staining of the physically larger compartments in lipid-treated cells is likely to reflect protein re-distribution within the membrane.

6/ Previously, authors reported that infection of RAW 264.7 MF with *M. bovis* BCG causes a significant upregulation of NPC1 protein; however, this is not clearly observed when using lipids from different *M. tuberculosis* strains, where a sustained exposure to lipids may be required. How realistic is this? Little is known about the biosynthesis and restructuration of the *M. tuberculosis* cell wall during infection, which could be also host cell dependent.

Our previous publication⁸ demonstrated increased expression of NPC1 when macrophages were infected with live BCG. However, in the study made here cells were exposed only to mycobacterial lipid extracts. Therefore, it is difficult to make a direct comparison between the effects because of the different experimental approaches. We have therefore modified the text:

“Currently, we are unable to fully explain the differences between these data, but one possibility is that infection with live mycobacteria achieves greater inhibition of NPC1 activity and as a consequence an increase in protein abundance in an attempt to compensate for NPC1 loss of function”.

7/ It seems that NPC1 is not the full story, may be part of the story, and thus better correlations need to be presented. For example, is there a correlation between *M. tuberculosis* intracellular growth of the different strains studied and the levels of inhibition for NPC1?

Studies involving infection of cells with live *Mtb* strains were beyond the scope of this manuscript. In addition, as we have now made clearer that there is currently no assay to measure directly NPC1 function. Therefore, such studies, although of interest, would not be straightforward to undertake and analyze.

8/ Authors need to consider the existence of other cell wall components, such as lipoglycans and glycosylated-lipoproteins that are not being studied here and also directly contribute to *M. tuberculosis* survival within host cells by inducing phagosome maturation arrest, this is not being studied, either mentioned. In addition, some of the major regulators of the faith of *M.tb* within macrophages are the mannose-containing lipoglycans. How the presence of these compounds mask/enhance the effects of the lipids as shown here?

We thank the reviewer for bringing to our attention these *Mtb* components. In fact, we had assayed and found a lack of activity in several *Mtb* cell wall components and have now included these data (Fig S2B).

We cannot rule out that other cell envelope components such as lipoglycans or lipoproteins could have an impact on the phenotype observed using whole bacteria and not total lipid extracts like in this study. As reported in Figure S3B, ManLAM did not induce an increase in LysoTrackerTM MFI which agrees with our reported phenotype caused by extractible lipids. In addition, in the total lipid extracts generated, it is quite unlikely LAM could be present as its extraction requires a dedicated protocol³¹⁻³³.

9/ It is not clear why other lineages, especially lineage 3 was not studied, especially in the light that strain in lineage 3 are causing drug resistance TB outbreaks.

Our intention was to focus on a collection of *Mtb* strains from different lineages that have been well characterised³⁴. However, in response to this request we have obtained a lipid extract from a strain from lineage 3 (the only one currently available from BEI) and have now included new data showing that it also possesses NPC1 inhibitory activity (Fig 2C).

10/ *There are some overstatements generalizing the results, when authors only studied few M.tb strains in the NPC pathway experiments. The use of NTM and M. smegmatis, as well as other M. tuberculosis strains lipids, was only performed in the acidification study.*

In our previous publication⁸ we demonstrated that *M. smegmatis* did not induce any of the NPC cellular phenotypes, including cholesterol accumulation, increased LysoTracker™ staining, altered trafficking of GM1, sphingomyelin storage and GSL and sphingosine accumulation.

We used the intensity of LysoTracker™ staining as an initial and quantitative assay to screen all of the *Mtb* strains and other mycobacteria. Because of the lower throughput of all of the different assays used to confirm the induction of the multiple cellular phenotypes that collectively define NPC, we focused efforts on strains representative of the major *Mtb* lineages that were available to us.

For clarification, LysoTracker™ staining is not applicable for the measurement of acidification of the lysosome. LysoTracker™ intensity provides a measure of the relative volume of the acidic compartment of the cell⁹

11/ *There is not a direct read out of NPC1 activity being diminished. Studies using the inhibitor plus the lipids will add a light into demonstrating if lipids further enhance NPC1 inhibition or using NPC1 activity stimulants plus the mycobacterial lipids studied.*

Unfortunately, there is no assay that can directly measure NPC1 activity and therefore the extent of inhibition. We therefore examined for the induction of cellular phenotypes that are known to result from loss of NPC1 activity. Previously, we have addressed this issue by comparing the relative capacity of pathogenic mycobacteria to induce NPC phenotypes in wild type cells and cells overexpressing NPC1⁸. We confirmed that *Mtb* lipid target NPC1 and that cells overexpressing the protein are more resistant to the induction of NPC phenotypes.

12/ *The extraction of lipids used will extract lipooligosaccharides (LOS, polar lipids); however, the resuspension of the extracted lipids with only chloroform will make difficult the analysis of LOS. In addition, the argument that M. smegmatis does not induce NPC1 because of the presence of differential lipids such as non-serovar specific glycopeptidolipids (GPLs) fails in concept, because strains of the M. avium complex seem to induce NPC1 and have both non-serovar and serovar specific GPLs.*

We thank the reviewer for this comment about that important detail concerning our procedure to extract lipids. It is true that polar lipids are extracted with our methods. Even though we used only chloroform to solubilise the final extract prior to TLC, we can clearly see from the TLCs presented in Figure S4A that lipids such as phosphatidyl-*myo*-inositol mannosides, which are highly polar glycolipids, are being detected and therefore solubilized. For the analysis of LOS, a similar procedure as ours, which consists in solubilising the dried total lipid extract by chloroform, is used^{30,35-37}.

Regarding the difference in response between *M. smegmatis* and *M. avium* is very pertinent and more efforts have to be conducted in order to decipher this discrepancy in NPC1 inhibition. Nevertheless, we can propose that some specific lipids or abundances of particular lipids must be taken into consideration in order to explain that observation.

We have now added the following statement:

“Regarding the phenotype observed with *M. smegmatis* extract, perhaps discrepancy in abundances of particular lipids or presence of a significant amount of exposed glycopeptidoglycans^{38,39} may be the cause of the non-induction of NPC phenotypes.”

13/ Fig. 2: Acidification induced by total lipids happens for lineage 4 (100% of the strains tested, 7/7), lineage 1 (60% of the strains tested (3/5)) and lineage 2 (67% of the strains tested (2/6) but with very heterogeneous MFIs values. Of these, authors only choose one strain from each lineage to determine the effects of the lipids on the NCP pathway vs. *M. canettii* and *M. tuberculosis* H37Rv. Although reasonable, this selection limits the impact of their claims; and thus, one cannot extrapolate their results to the general statements that authors mention in the results, discussion and abstract (as well as title).

First, we should clarify that the intensity of LysoTracker™ staining is a measure of the relative volume of acidic compartment of the cell and not pH.

As we have indicated above, we now include data confirming activity in all of the strains representative of lineages 1 and 2 and a strain from lineage 3 (Fig 1C). Because of the multiple assays required to confirm induction of NPC cellular phenotypes we focused efforts on strains representative of lineages 1, 2 and 4, along with *M. canettii*. We have modified the text to illustrate this.

14/ Fig 3: Need quantifications. By flow, UT, DMSO, *M. canettii* show similar results; however, by confocal looking at LAMP-1 these results differ, where *M. canettii* or DMSO seem higher than even the positive control U18666A. In this regard, the positive control appears similar to untreated cells. Further, when comparing conditions/strains with similar values [as obtained by flow (MFI)], in some cases these differ by confocal microscopy (e.g. UT vs. 639). Further, another marker of lysosome accumulation such as CD63 will be necessary to confirm results. As well as other type of cells. It is not clear how many times the experiment in Fig. 3 Panel A was done. Also, quantifications are required as it seems that DMSO produces higher accumulation of LAMP-1 than the positive control.

We have repeated these investigations and made detailed quantification to confirm the findings. We provide both lower and higher magnification of representative fields (Fig 2 and 3), along with quantification of mean lysosome size (diameter) and size distribution for each of the treatments. These data confirm a statistically significant increase in lysosomal volume in cells treated with U18666A and lipids from the *Mtb* strains examined, but not with lipid extract of *M. canettii* (Fig 2 and 3). Furthermore, we have provided additional evidence of increased lysosome volume by quantification of microscopy images of LysoTracker™ staining (Fig 2).

Lysosomal accumulation of cholesterol and GM1 are defining and universally-accepted cellular features of NPC that have been demonstrated multiple times in the published literature.

15/ Fig 4: Irrelevant if the impact is on NPC1 activity (acidification and accumulation of lysosomes) and not on NPC1 levels. Further, which *M. tuberculosis* H37Rv lipids are the ones tested here (BEI or in house obtained)? *M. canettii* in the blot presented in Fig. 4 looks less than DMSO but the quantification shows similar results. A better representative blot may be needed. Actually, the beta-actin in the blot for *M. canettii* seems higher so the value of the NPC1 should be event lower when normalized to beta-actin. It is not clear why the U18666A control is added, are the authors expecting a decrease in levels, activity or both? And if it is a decrease in activity, this is not directly well proven.

We should clarify that there is extensive evidence in the literature that drugs that block NPC1 activity leads to increased NPC1 synthesis^{1,2}.

In response to the reviewers comments we have prepared new blots and provide new quantification (Fig 4). Quantification confirmed a significant increase in Npc1 protein with U18666A and H37Rv lipid extract (Fig 4B), but an absence of a significant increase with the other treatments (Fig 4B)

16/ Fig. 5: Fig. 5A and B can be combined. It is not clear from the selected pics that *M. canettii* has less puncta than *M. tuberculosis* H37Rv. Some regions marked by arrows in *M. tuberculosis* H37Rv look similar to *M. canettii* but these are not marked. Authors need to clarify what they chose to mark or add representative images of their findings. Further, authors state that these are lysosomes containing cholesterol. Proper markers will need to be used to show co-localization of lysosomes with cholesterol. Graphs in 5Ci and 5Cii can be combined, the issue here is that any of the lipids extracts induce cholesterol accumulation at the levels observed by U18666A. This does not match the results obtained in 5B.

In order to achieve clearer illustration of filipin staining we have replaced fluorescent images with black and white images, as we have done previously³, We have provided an explanation of the method used to discriminate cholesterol puncta in materials and methods.

Accumulation of cholesterol in the endo-lysosome is characteristic of NPC (and was the clinical basis for diagnosis for several decades), is universally accepted and has been demonstrated in multiple publications^{1,40} therefore, we do not consider it essential to repeat this information. In the data sets U18666A treatment was employed as a positive control for induction of NPC cellular phenotypes. The increase in cholesterol puncta induced by *Mtb* lipids, although lower than with U18666A are all statistically significant as compared to control. The larger increase in signal seen with U18666A likely reflects greater inhibition because of the drug being a pure NPC1 antagonist and/or increased availability because the drug is itself lysosomotropic.

It is not clear if this is lysosomal or endosomal accumulation, this needs to be discerned to make this claim. Co-localization of Filipin with LAMP-1 or CD63 will be required. It will be of interest to show if the accumulation of LDL-vesicles also occurs as shown for NPC, and what this will mean.

Please see response to previous comment – endo-lysosomal accumulation of cholesterol is a universally accepted cellular phenotype of NPC and has been documented on multiple occasions.

17/ Fig 6: Representative figures need to be chosen carefully. As in previous figures, *M. canettii* seems to have more puncta than *M. tuberculosis* 333. Images provided seem that do not represent the quantification presented. In addition, authors need to revise the scale, because although in the figure the scale is the same, cells in *M. canettii* seem (perception) smaller than in the other groups. Also authors need to verify that accumulation happened in lysosomes vs. early/late endosomes.

We can confirm that the magnifications and scale bars for all of the images are the same and correct. Interestingly, we frequently observed a reduction in size of cells treated with *M. canettii* extract (see Figs 2, 3, 5 and 6), but we are unsure of its biological relevance.

18/ Fig 7: There are some quantifications that show significance in GM populations after exposure to the lipids of some *M. tuberculosis* strains vs. *M. canettii* but the HPLC plots as presented do not show these significant differences. For comparative reason, graphs in Fig. 7C need to be in the same scale.

The single HPLC profiles shown are representative and as such cannot simultaneously easily visualise differences in the peaks of both high and low abundance GSLs. The quantification of each GSL is based upon careful inspection of each species individually from the HPLC traces and comparison with standards. The scale of HPLC trace for *M. canettii* has been replaced so as to be identical to the others (Fig 7).

19/ Fig S3: This figure needs to be labeled and explained better. In panel E some strains in this panel are irrelevant to the presented study, and add confusion showing not clear lipid differences between strains that cause or do not cause the NPC1 phenotype; thus, without defining an obvious link.

We have now added labels and amended the figure legend.

20/ Fig S3A: There are differences in the levels of some of the lipids on the strains evaluated in this study. For example, strain 333 has less DIM B but more TAG than the other 2 strains (281 and 639). Same for some of the PIMs and TMM.

21/ Fig S3B: Again, differences in lipids are shown in supplemental Figure S3. Correlations between the presence/abundance of a specific lipids and the phenotypes observed will be important. This study is also no accounting the expression of these lipids during infection as well as other cell wall components present in the cell envelope that are not in the lipid fraction.

It is true that there are discreet differences in the abundances of some the lipids between the strains used in this study. Nevertheless, here we focused on the general response of total lipid extracts rather than identifying a particular and specific lipid that cause NPC1 inhibition. We are definitively eager to identify that or those lipids that induce NPC1 but that is beyond the scope of the current study that reports on the evolution of MTB complex and its ability to inhibition NPC1 through extractible lipids.

In addition, a difference between *M. canettii* and other MTBC strains is the presence of phenolic glycolipid, present in HN878 and *M. bovis*. This could be experimentally addressed.

We thank the reviewer for this comment about the possibility that phenolic glycolipid could induce NPC1. As mentioned previously, in this report we wanted to understand the evolution of MTB through the ability of total lipid extracts to induce NPC1. We cannot rule out the possibility that PGL could account for this phenotype, but this is beyond the scope of the current study.

22/ Dispersed soluble lipids vs. in micellar form, did the authors try different amounts of lipids to see if the phenomena of micelle formation plays a role? is there a total lipid dose response?

It is not straightforward to address this specific question when only lipid extracts of mixed composition are available. We have included data to confirm that the extent of LysoTracker™ intensity is dose-dependent (Fig S2).

Minor:

23/ Fig. 1 could be moved to supplement, and the phylogenetic tree used based on the expression of *HupB* needs to be better justified in the introduction.

As requested, we have moved Figure 1 to Supplementary Figures. This figure⁴¹ was chosen purely for illustrative purposes as a phylogenetic tree showing an overview of the genus showing some of the organisms used in this study. No assumptions are made about the importance, or not of *HupB*. If the reviewer knows of a more appropriate image, we would be pleased to consider it.

24/ In some experiments/figures it is not clear how many cells/events were counted.

We have included this information as it was inadvertently not included.

25/ "TB is a pathogen". This is inaccurate as stated and needs to be revised.

This inadvertent error has been corrected.

26/ Table 2: Mean +/- SEM needs to be presented in this table.

This information is now included in Table 2.

References

- 1 Vanier, M. T. Niemann-Pick disease type C. *Orphanet J Rare Dis* **5**, 16, doi:10.1186 (2010).
- 2 Watari, H. *et al.* Determinants of NPC1 expression and action: key promoter regions, posttranscriptional control, and the importance of a "cysteine-rich" loop. *Exp Cell Res* **259**, 247-256 (2000).
- 3 Hoglinger, D. *et al.* NPC1 regulates ER contacts with endocytic organelles to mediate cholesterol egress. *Nat Commun* **10**, 4276, doi:10.1038 (2019).
- 4 Vanier, M. T. *et al.* Type C Niemann-Pick disease: biochemical aspects and phenotypic heterogeneity. *Dev Neurosci* **13**, 307-314 (1991).
- 5 Morris, J. A. & Carstea, E. D. Niemann-Pick C disease: cholesterol handling gone awry. *Mol Med Today* **4**, 525-531 (1998).
- 6 Fog, C. K. & Kirkegaard, T. Animal models for Niemann-Pick type C: implications for drug discovery & development. *Expert Opin Drug Discov* **14**, 499-509 (2019).
- 7 Lloyd-Evans, E. *et al.* Niemann-Pick disease type C1 is a sphingosine storage disease that causes deregulation of lysosomal calcium. *Nat Med* **14**, 1247-1255, (2008).
- 8 Fineran, P. *et al.* Pathogenic mycobacteria achieve cellular persistence by inhibiting the Niemann-Pick Type C disease cellular pathway. *Wellcome Open Res* **1**, 18, doi:10.12688 (2016).
- 9 te Vrugte, D. *et al.* Relative acidic compartment volume as a lysosomal storage disorder-associated biomarker. *J Clin Invest* **124**, 1320-1328 (2014).
- 10 Sezgin, E. *et al.* A comparative study on fluorescent cholesterol analogs as versatile cellular reporters. *J Lipid Res* **57**, 299-309 (2016).
- 11 Lu, F. *et al.* Identification of NPC1 as the target of U18666A, an inhibitor of lysosomal cholesterol export and Ebola infection. *Elife* **4**, doi:10.7554 (2015).
- 12 Ernst, J. D., Trevejo-Nunez, G. & Banaiee, N. Genomics and the evolution, pathogenesis, and diagnosis of tuberculosis. *J Clin Invest* **117**, 1738-1745, (2007).

- 13 Supply, P. & Brosch, R. The Biology and Epidemiology of *Mycobacterium canettii*. *Adv Exp Med Biol* **1019**, 27-41 (2017).
- 14 Bottai, D. *et al.* TbD1 deletion as a driver of the evolutionary success of modern epidemic *Mycobacterium tuberculosis* lineages. *Nat Commun* **11**, 684, doi:10.1038 (2020).
- 15 Orgeur, M. & Brosch, R. Evolution of virulence in the *Mycobacterium tuberculosis* complex. *Curr Opin Microbiol* **41**, 68-75 (2018).
- 16 Jackson, M. The mycobacterial cell envelope-lipids. *Cold Spring Harb Perspect Med* **4**, doi:10.1101 (2014).
- 17 Boritsch, E. C. *et al.* pks5-recombination-mediated surface remodelling in *Mycobacterium tuberculosis* emergence. *Nat Microbiol* **1**, 15019, doi:10.1038 (2016).
- 18 Howard, S. T. *et al.* Spontaneous reversion of *Mycobacterium abscessus* from a smooth to a rough morphotype is associated with reduced expression of glycopeptidolipid and reacquisition of an invasive phenotype. *Microbiology (Reading)* **152**, 1581-1590 (2006).
- 19 Byrd, T. F. & Lyons, C. R. Preliminary characterization of a *Mycobacterium abscessus* mutant in human and murine models of infection. *Infect Immun* **67**, 4700-4707, (1999).
- 20 Ripoll, F. *et al.* Genomics of glycopeptidolipid biosynthesis in *Mycobacterium abscessus* and *M. chelonae*. *BMC Genomics* **8**, 114, doi:10.1186 (2007).
- 21 Catherinot, E. *et al.* Hypervirulence of a rough variant of the *Mycobacterium abscessus* type strain. *Infect Immun* **75**, 1055-1058 (2007).
- 22 Jackson, M., Stevens, C. M., Zhang, L., Zgurskaya, H. I. & Niederweis, M. Transporters Involved in the Biogenesis and Functionalization of the Mycobacterial Cell Envelope. *Chem Rev* **121**, 5124-5157 (2021).
- 23 Szewczyk, R., Kowalski, K., Janiszewska-Drobinska, B. & Druszczynska, M. Rapid method for *Mycobacterium tuberculosis* identification using electrospray ionization tandem mass spectrometry analysis of mycolic acids. *Diagn Microbiol Infect Dis* **76**, 298-305 (2013).
- 24 Marrakchi, H., Laneelle, M. A. & Daffe, M. Mycolic acids: structures, biosynthesis, and beyond. *Chem Biol* **21**, 67-85 (2014).
- 25 Song, S. H. *et al.* Electrospray ionization-tandem mass spectrometry analysis of the mycolic acid profiles for the identification of common clinical isolates of mycobacterial species. *J Microbiol Methods* **77**, 165-177 (2009).
- 26 Hatzios, S. K. *et al.* PapA3 is an acyltransferase required for polyacyltrehalose biosynthesis in *Mycobacterium tuberculosis*. *J Biol Chem* **284**, 12745-12751 (2009).
- 27 Layre, E. *et al.* Deciphering sulfoglycolipids of *Mycobacterium tuberculosis*. *J Lipid Res* **52**, 1098-1110 (2011).
- 28 Burbaud, S. *et al.* Trehalose Polyphleates Are Produced by a Glycolipid Biosynthetic Pathway Conserved across Phylogenetically Distant Mycobacteria. *Cell Chem Biol* **23**, 278-289 (2016).
- 29 Layre, E. *et al.* Molecular profiling of *Mycobacterium tuberculosis* identifies tuberculosinyl nucleoside products of the virulence-associated enzyme Rv3378c. *Proc Natl Acad Sci USA* **111**, 2978-2983 (2014).
- 30 Malaga, W. *et al.* Deciphering the genetic bases of the structural diversity of phenolic glycolipids in strains of the *Mycobacterium tuberculosis* complex. *J Biol Chem* **283**, 15177-15184 (2008).

- 31 Nigou, J. *et al.* The phosphatidyl-myo-inositol anchor of the lipoarabinomannans from *Mycobacterium bovis* bacillus Calmette Guerin. Heterogeneity, structure, and role in the regulation of cytokine secretion. *J Biol Chem* **272**, 23094-23103 (1997).
- 32 Nigou, J., Gilleron, M. & Puzo, G. Lipoarabinomannans: characterization of the multiacylated forms of the phosphatidyl-myo-inositol anchor by NMR spectroscopy. *Biochem J* **337 (Pt 3)**, 453-460 (1999).
- 33 Nigou, J., Gilleron, M., Brando, T. & Puzo, G. Structural analysis of mycobacterial lipoglycans. *Appl Biochem Biotechnol* **118**, 253-267 (2004).
- 34 Krishnan, N. *et al.* *Mycobacterium tuberculosis* lineage influences innate immune response and virulence and is associated with distinct cell envelope lipid profiles. *PLoS One* **6**, e23870, doi:10.1371 (2011).
- 35 Simeone, R. *et al.* Delineation of the roles of FadD22, FadD26 and FadD29 in the biosynthesis of phthiocerol dimycocerosates and related compounds in *Mycobacterium tuberculosis*. *FEBS J* **277**, 2715-2725 (2010).
- 36 Ren, H. *et al.* Identification of the lipooligosaccharide biosynthetic gene cluster from *Mycobacterium marinum*. *Mol Microbiol* **63**, 1345-1359 (2007).
- 37 Burguiere, A. *et al.* LosA, a key glycosyltransferase involved in the biosynthesis of a novel family of glycosylated acyltrehalose lipooligosaccharides from *Mycobacterium marinum*. *J Biol Chem* **280**, 42124-42133 (2005).
- 38 Villeneuve, C. *et al.* Surface-exposed glycopeptidolipids of *Mycobacterium smegmatis* specifically inhibit the phagocytosis of mycobacteria by human macrophages. Identification of a novel family of glycopeptidolipids. *J Biol Chem* **278**, 51291-51300 (2003).
- 39 Villeneuve, C. *et al.* Mycobacteria use their surface-exposed glycolipids to infect human macrophages through a receptor-dependent process. *J Lipid Res* **46**, 475-483 (2005).
- 40 Millat, G. *et al.* Niemann-Pick C1 disease: the I1061T substitution is a frequent mutant allele in patients of Western European descent and correlates with a classic juvenile phenotype. *Am J Hum Genet* **65**, 1321-1329 (1999).
- 41 Pandey, S. D. *et al.* Iron-regulated protein HupB of *Mycobacterium tuberculosis* positively regulates siderophore biosynthesis and is essential for growth in macrophages. *J Bacteriol* **196**, 1853-1865 (2014).

REVIEWERS' COMMENTS

Reviewer #2 (Remarks to the Author):

In the revised version of their manuscript NCOMMS-21-49975, authors addressed all questions I raised in my first review. Moreover, more details, explanations and new data have been added to better clarify their results. Figures have been corrected.

From my point of view, the paper can now be accepted for publication.

Reviewer #3 (Remarks to the Author):

Authors somehow addressed my previous concerns, some of them still stand, but the authors indicated that these will take some time and it is an ongoing effort in their lab (e.g. identify the lipid/s driving the phenotype).

Minor:

Change glycopeptidogycans to glycopeptidolipids, these are not the same.

Response to Reviewers Comments

REVIEWERS' COMMENTS

Reviewer #2 (Remarks to the Author):

In the revised version of their manuscript NCOMMS-21-49975, authors addressed all questions I raised in my first review.

Moreover, more details, explanations and new data have been added to better clarify their results. Figures have been corrected.

From my point of view, the paper can now be accepted for publication.

As requested by the reviewer, we have not made any changes to the manuscript.

Reviewer #3 (Remarks to the Author):

Authors somehow addressed my previous concerns, some of them still stand, but the authors indicated that these will take some time and it is an ongoing effort in their lab (e.g. identify the lipid/s driving the phenotype).

Minor:

Change glycopeptidogycans to glycopeptidolipids, these are not the same.

As requested by this reviewer, we have changed the terminology from glycopeptidogycans to glycopeptidolipids (page 18 of manuscript)